# Rationally derived inhibitors of hepatitis C virus (HCV) p7 channel activity reveal prospect for bimodal antiviral therapy

Joseph Shaw[1,2], Rajendra Gosain[2,3], Monoj Mon Kalita[4], Toshana L Foster[1,2†], Jayakanth Kankanala[2,3‡], D Ram Mahato[4], Sonia Abas[2,3], Barnabas J King[5], Claire Scott[1,2§], Emma Brown[1,2], Matthew J Bentham[1,2], Laura Wetherill[1,2], Abigail Bloy[1,2], Adel Samson[1], Mark Harris[2,6], Jamel Mankouri[2,6], David J Rowlands[2,6], Andrew Macdonald[2,6], Alexander W Tarr[5], Wolfgang B Fischer[4], Richard Foster[2,3*], Stephen Griffin[1,2*]

[1]Leeds Institute of Medical Research, School of Medicine, Faculty of Medicine and Health, University of Leeds, St James' University Hospital, Leeds, United Kingdom; [2]Astbury Centre for Structural Molecular Biology, University of Leeds, Woodhouse Lane, Leeds, United Kingdom; [3]School of Chemistry, University of Leeds, Woodhouse Lane, Leeds, United Kingdom; [4]Institute of Biophotonics, National Yang-Ming University, Taipei, Taiwan; [5]School of Life Sciences, Faculty of Medicine & Health Sciences, University of Nottingham, Queen's Medical Centre, Nottingham, United Kingdom; [6]School of Molecular & Cellular Biology, Faculty of Biological Sciences, University of Leeds, Woodhouse Lane, Leeds, United Kingdom

**\*For correspondence:**
r.foster@leeds.ac.uk (RF);
s.d.c.griffin@leeds.ac.uk (SG)

**Present address:** [†]School of Veterinary Medicine and Science, Faculty of Medicine & Health Sciences, University of Nottingham, Sutton Bonington Campus, Leicestershire, United Kingdom; [‡]Center for Drug Design, University of Minnesota Twin Cities, Minneapolis, United States; [§]Covance Clinical Research Unit Ltd, Springfield House, Leeds, United Kingdom

**Competing interests:** The authors declare that no competing interests exist.

**Abstract** Since the 1960s, a single class of agent has been licensed targeting virus-encoded ion channels, or 'viroporins', contrasting the success of channel blocking drugs in other areas of medicine. Although resistance arose to these prototypic adamantane inhibitors of the influenza A virus (IAV) M2 proton channel, a growing number of clinically and economically important viruses are now recognised to encode essential viroporins providing potential targets for modern drug discovery. We describe the first rationally designed viroporin inhibitor with a comprehensive structure-activity relationship (SAR). This step-change in understanding not only revealed a second biological function for the p7 viroporin from hepatitis C virus (HCV) during virus entry, but also enabled the synthesis of a labelled tool compound that retained biological activity. Hence, p7 inhibitors (p7i) represent a unique class of HCV antiviral targeting both the spread and establishment of infection, as well as a precedent for future viroporin-targeted drug discovery.

## Introduction

Hepatitis C virus (HCV) represents a global clinical challenge as a major cause of chronic liver disease, with severe complications including cirrhosis, liver failure and primary liver cancers (hepatocellular- and intrahepatic cholangio- carcinomas (HCC, iCCA)). Acute infection is predominantly asymptomatic which, combined with limited awareness and population screening, means that liver disease is often advanced upon diagnosis. WHO estimates put the total number of deaths due to HCV infection in 2015 at more than 400 000, with ~1.75 million new infections annually.

HCV antiviral therapy, originally comprising recombinant type one interferon (IFN) combined with the guanosine analogue ribavirin, has been revolutionised by new direct-acting antivirals (DAA). DAA are an unprecedented drug development success, capable of achieving high rates of cure with favourable toxicity profiles enabling their use in patients with advanced disease (**Baumert et al.,**

*2019*). Current DAA target three proteins within the viral replicase (NS3/4A protease, NS5A and the NS5B RNA-dependent RNA polymerase (RdRP)), with drug combinations available for treating each of the eight viral genotypes.

However, the absence of an HCV vaccine, or other means of prophylaxis, makes DAA-based eradication strategies proposed for ~71 million chronically infected individuals immensely challenging. DAA availability remains limited by cost, coincident with poor diagnostic rates and rapidly increasing burden in low/middle-income countries (LMIC). Resistant viral variants are an increasing concern (*Pawlotsky, 2016*), with recent reports of increased resistance amongst some rarer viral subtypes (*Fourati et al., 2019*). Compliance within high-risk populations is low and successful DAA therapy does not prevent re-infection. Moreover, recent studies suggest that DAA are less able to reduce the risk of HCC in treated patients compared with IFN-based therapy (*Baumert and Hoshida, 2019*). This may be linked to virus-induced host epigenetic signatures that are not reversed following DAA cure (*Hamdane et al., 2019*; *Perez et al., 2019*).

HCV is an enveloped positive sense RNA virus with a ~ 9.6 kb genome encoding a single large polyprotein translated from an internal ribosomal entry site (IRES) in the 5'-untranslated region. The polyprotein is spatially organised into structural components at the amino terminus and replicase proteins towards the carboxyl terminus; these are released by host and viral proteases, respectively. In addition, p7 and NS2 play pivotal roles during virion assembly (*Gentzsch et al., 2013*) involving protein–protein interactions with one another, as well as other viral proteins (*Jirasko et al., 2008*; *Jirasko et al., 2010*; *Boson et al., 2011*). Furthermore, the 63 amino acid p7 protein is capable of oligomerising (forming hexamers and/or heptamers *Luik et al., 2009*; *Clarke et al., 2006*) within membranes to form an ion channel complex (*Breitinger et al., 2016*; *Premkumar et al., 2004*; *Pavlović et al., 2003*; *Griffin et al., 2003*) with a distinct, but equally essential role during virion secretion (*Steinmann et al., 2007a*; *Jones et al., 2007*). This comprises the raising of secretory vesicle pH, which is necessary to protect acid-labile intracellular virions (*Atkins et al., 2014*; *Bentham et al., 2013*; *Wozniak et al., 2010*).

Prototypic compounds, such as adamantanes and alkyl imino-sugars, inhibit p7 channel activity as well as virion secretion in culture, but with relatively poor potency (*Griffin et al., 2008*; *Steinmann et al., 2007b*). However, identification of explicit resistance polymorphisms confirmed that such effects are specific (*Foster et al., 2011*). This includes Leu20Phe, which confers resistance in genotype 1b and 2a p7 to adamantanes, including rimantadine. Thus, despite poor potency, prototypic inhibitors highlight druggable regions upon p7 channel complexes suited to targeting by improved compounds.

The majority of p7 structural studies support the folding of protomers into a hairpin conformation (*Luik et al., 2009*; *Foster et al., 2014*; *Cook et al., 2013*; *Montserret et al., 2010*). This is in agreement with immuno-gold labelling of p7 channel complexes by electron microscopy (EM)(*Luik et al., 2009*), immunofluorescence studies of epitope-tagged p7 expressed in mammalian cells (*Carrère-Kremer et al., 2002*), and the membrane topology necessary to orient NS2 correctly within the ER membrane during translation of the viral polyprotein. Resultant p7 channel models comprise hexa- or heptameric assemblies of tilted protomers and a lumen formed by the N-terminal helix containing a well conserved (~90%) His17 residue, as proven biochemically (*Chew et al., 2009*). However, a solution NMR structure of a complete hexameric p7 channel complex comprised protomers in an unusual intertwined triple-helix configuration (PDB: 2M6X) (*OuYang et al., 2013*). This structure retained a wider channel lumen compared to hairpin-based structures and exposed conserved basic residues to the lipid bilayer. Functionality of this sequence was not demonstrable, possibly due to mutagenesis of conserved cysteine residues to enhance recombinant expression. Furthermore, recent studies have questioned the validity of this structure due to potential artefacts caused by alkyl-phosphocholine detergents used as membrane mimetics *Oestringer et al., 2018*; the original authors contest this notion (*Chen et al., 2018*). However, molecular dynamics simulations favour the stability and channel gating characteristics of hairpin-based structures (*Chandler et al., 2012*; *Holzmann et al., 2016*).

Interestingly, both hairpin- and triple-helix-based channel structures retain an adamantane- binding site upon the channel periphery that includes position 20 (*Foster et al., 2014*; *OuYang et al., 2013*). Unsurprisingly, the conformation and amino acid content of this site differs significantly between structures. Previously, we used a hairpin-based heptameric channel complex as a template for in silico high throughput screening, based upon a genotype 1b monomeric hairpin p7 solution

NMR structure (PDB: 3ZD0). Resultant chemical hits displayed considerably improved potency compared with rimantadine that was independent of Leu20Phe mutations (*Foster et al., 2014*). However, initial hits lacked convergence around a common pharmacophore and this prevented understanding of a structure-activity relationship (SAR).

We now present a second-generation lead-like oxindole based inhibitor of p7 channel activity complete with a comprehensive SAR: 'JK3/32'. The resultant step forward in potency and specificity has not only led to the identification of a second biological role for p7 channel activity during virus entry, but also enabled the generation of a fluorescently labelled tool compound that retained biological activity. This distinguishes p7 inhibitors (p7i) from other DAAs by targeting two discrete stages of the virus life cycle separate to genome replication and sets a new precedent for viroporin-targeted drug design.

## Results

### Refining a rapid throughput assay for secreted HCV infectivity

p7 channel activity is essential for the secretion of infectious virions (*Wozniak et al., 2010*), making secreted infectivity an ideal biomarker readout for inhibitor antiviral effects. To expedite testing of secreted HCV infectivity following treatment with high numbers of compounds and/or concentration points, we adapted previously published protocols using the IncuCyte ZOOM (*Stewart et al., 2015*) to quantify infection of naïve cells immunostained for NS5A (*Figure 1A*). Dilution of virus-containing supernatants was optimised (1:4) for signal-to-noise whilst accurately reflecting infectivity (i.e. within the linear range of a dilution series correctly determining virus titre, see *Figure 1—figure supplement 1A*). This negates the need for serial dilutions and removes both human error and the amplification of said error due to multiplication by large dilution factors. The assay is applicable to multiple native or chimeric viruses and readily generates 8-point $EC_{50}$ curves in a reporter-free system.

Treatment with the RdRP inhibitor, sofosbuvir (SOF), reduced viral replication within transfected producer cells (*Figure 1—figure supplement 1B*), and this was previously shown to be directly proportional to secreted infectivity (*Stewart et al., 2015*). Note, the use of RNA transfected producer cells permits the study of virus secretion in the absence of potentially confounding effects upon virus entry. The NS5A inhibitor, daclatasvir (DCV) also reduced secreted infectivity, with no effect upon cellular toxicity monitored by producer cell confluency across 8-point dilution ranges (*Figure 1—figure supplement 1C*). The ability of the assay to identify false positives caused by cellular toxicity was confirmed using a HSP90 inhibitor, radicicol, which caused a reduction in secreted infectious particles congruent with producer cell viability (note, cell viability was confirmed by parallel MTT assays, see Materials and methods). In addition to detecting anti-viral activity of replication inhibitors SOF and DCV, the assay successfully quantified anti-viral effects of the prototypic p7i, rimantadine, confirming the ability to monitor effects upon p7-dependent virus secretion (*Figure 1B*). The assay Z factor was determined as 0.47 ± 0.14, with a % coefficient variation of 16.9 ± 4.3 and a signal/background ratio of 81.1 ± 23.6 (data from two independent positive and negative controls (SOF, DCV, DMSO only), in duplicate, averaged over three independent experimental repeats, with three separate assay plates, over two separate days).

### Identification of lead compound JK3/32

SAR-focused chemical modification (*Table 1*) of an oxindole core scaffold identified 'JK3/32' as a series lead with excellent potency against chimaeric genotype 1b HCV (J4/JFH-1) secretion ($EC_{50}$ ~184 nM) (*Figure 1B*, *Figure 1—figure supplement 1C*). Thus, JK3/32 potency was comparable to SOF and considerably improved compared with the prototypic p7i, rimantadine. As seen for first generation compounds (*Foster et al., 2014*), JK3/32 retained cross-genotype activity versus HCV genotype 3a ($EC_{50}$ ~738 nM), with a modest reduction in activity against more genetically distant 2a viruses ($EC_{50}$ ~1900 nM) (*Figure 1C*). The compound showed a toxicity-based selectivity index of >500 ($CC_{50}$ >100 000 nM) based upon confluency (*Figure 1—figure supplement 1C*) and MTT assay (*Figure 1—figure supplement 2A*) in Huh7 cells, and had no discernible effect upon replication of HCV JFH-1 subgenomic replicons, which retain the same replicase as chimaeric viruses but lack the structural proteins, p7 and NS2 (*Figure 1—figure supplement 2B*).

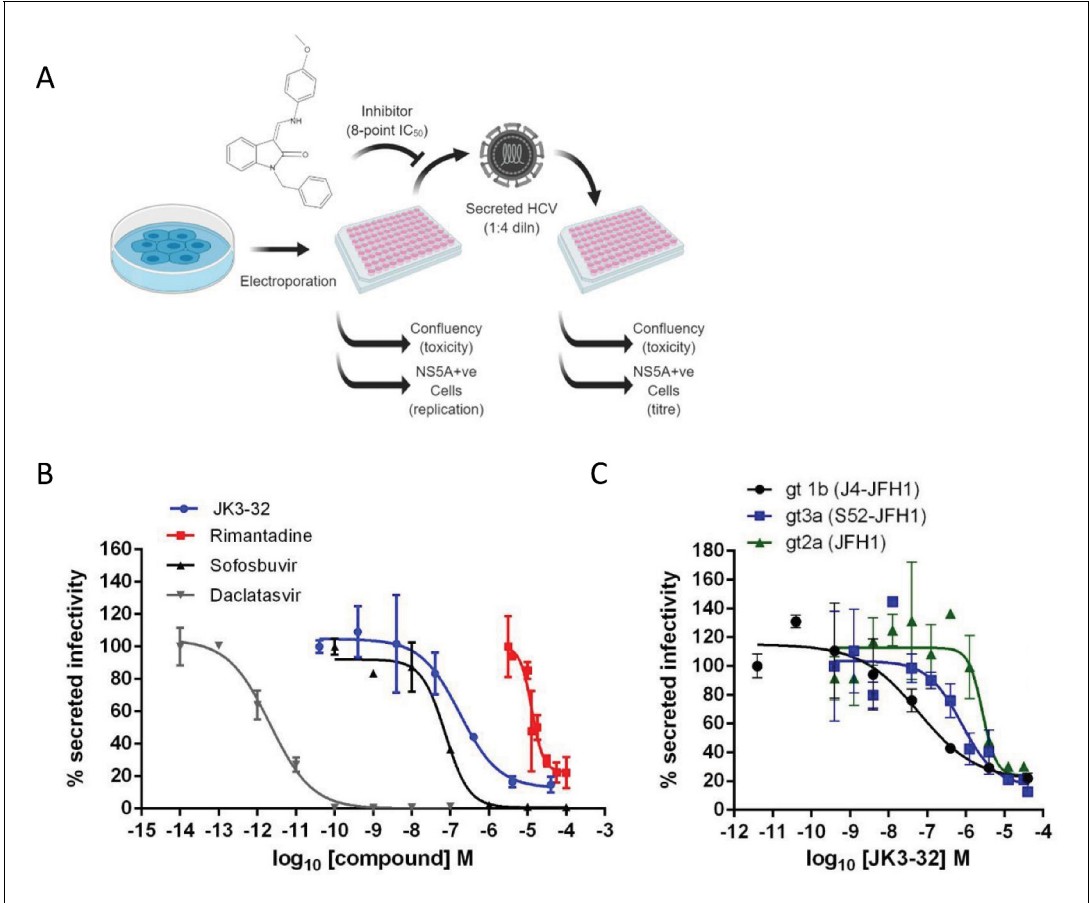

**Figure 1.** Activity of JK3/32 against HCV particle secretion. (A) Diagram of workflow for rapid throughput assay for secreted infectivity (generated using 'Biorender', https://app.biorender.com). (B) Comparison of JK3/32 potency vs. virion secretion of J4/JFH-1 with licensed HCV DAAs sofosbuvir and daclatasvir, as well as the prototypic adamantane viroporin inhibitor, rimantadine. Curves are representative of at least four experimental repeats for JK3/32, multiple for sofosbuvir and daclatasvir, and two for rimantadine, where each condition is carried out in quadruplicate and error bars represent standard deviations. (C) Comparative $EC_{50}$ curves for JK3/32 effects upon GT1b, 2a and 3a chimaeric HCV (J4, JFH-1, S52/JFH-1) secreted infectivity post-electroporation. Curves are again representative of multiple experiments and error bars show standard deviations between quadruplicate repeats. The online version of this article includes the following source data and figure supplement(s) for figure 1:

**Source data 1.** Raw data from EC50 curves.
**Figure supplement 1.** Optimised quantitation of secreted HCV infectivity using the IncuCyte Zoom.
**Figure supplement 1—source data 1.** Raw data for EC50 titrations.
**Figure supplement 2.** Screening for potential JK3/32 off-target effects.
**Figure supplement 2—source data 1.** Raw data for replicon and cyclosporine validations.
**Figure supplement 3.** Evolution of LDS19 compound series including JK3/32.
**Figure supplement 4.** In vitro dye release assay screen for activity of RS compounds versus genotype 1b p7.
**Figure supplement 4—source data 1.** Raw data for RS series iteration dye release assays.
**Figure supplement 5.** Effects of clinically advanced viroporin inhibitor, BIT225, versus particle secretion.
**Figure supplement 5—source data 1.** Raw data for BIT225 EC50 curve.

The oxindole scaffold of JK3/32 resembles that of certain licensed kinase inhibitor drugs (Sunitinib, Nintedanib). However, the JK series of inhibitors is chemically distinguished from these compounds by an *N*-alkyl substituent, which was essential for anti-HCV activity (*Table 1*); accordingly, JK3/32 displayed no off-target activity against a panel of human kinases tested commercially (*Figure 1—figure supplement 2C*).

JK3/32 was part of a chemical series derived through evolution of an original hit compound, LDS19, selected in silico using the 3ZD0 (monomeric) NMR structure to model a heptameric template (*Foster et al., 2014*; *Figure 1—figure supplement 3*). The first iteration of compounds

**Table 1.** SAR table for JK3/32 series showing compounds contributing directly.

Core oxindole scaffold for JK3/32 shown, indicating six R-groups (R[1]-R[6]) subjected to modification, as well as a position within the core structure (A).

**Activity vs gt1b p7 (J4-JFH1 HCV), 72 hr treatment**

| Compound | R[1] | R[2] | R[4] | R[5] | R[6] | A | EC$_{50}$ (µM) | CC$_{50}$ (µM) |
|---|---|---|---|---|---|---|---|---|
| JK3-32 | Bzl | H | OMe | H | H | CH | 0.184 ± 0.089 (n = 4) | >100 |
| JK3-42 | Ph | H | OMe | H | H | CH | 0.932 ± 0.812 (n = 2) | >4 |
| JK3-38 | H | H | OMe | H | H | CH | >4 (n = 2) | >4 |
| 21-RS-7 | Bzl | H | OMe | H | H | N | 1.34 ± 0.37 (n = 3) | >4 |
| 21-RS-8 | Bzl | H | H | H | H | N | 0.775 ± 0.700 (n = 2) | >40 |
| 21-RS-9 | Ph | H | OMe | H | H | N | >40 (n = 1) | >40 |
| 1191–104 | Bzl | OMe | H | H | H | CH | 11.33 (n = 1) | >40 |
| 1191–112 | Bzl | H | CN | H | H | CH | >4 (n = 1) | 13.3 ± 12.4 (n = 2) |
| 1191–121 | Bzl | H | OMe | H | F | CH | 0.40 ± 0.12 (n = 2) | 12.8 |
| 1191–120 | (4-CN-benzyl) | H | OMe | H | H | CH | 1.42 ± 1.38 (n = 2) | 5.4 ± 4.4 (n = 2) |
| 1191–124 | (4-F-benzyl) | H | OMe | F | H | CH | 0.32 ± 0.35 (n = 2) | 8.3 ± 3.9 (n = 2) |
| 1191–106 | (3,5-dimethylisoxazolyl-methyl) | H | OMe | H | H | CH | 11.58 (n = 1) | >40 |
| 1191–137 | (phenethyl) | H | OMe | H | H | CH | 2.30 (n = 1) | 12.5 |
| 1191–140 | (4-F-benzyl) | H | OMe | H | H | CH | 0.461 (n = 1) | 10.4 |
| 1191–141 | (4-F-benzyl) | H | F | H | H | CH | >1.26 | >1.26 |
| 1191–146 | (4-F-benzyl) | H | O-propargyl | H | H | CH | 2.48 (n = 1) | >12.6 |

*Table 1 continued on next page*

| 1191–125 |  | H | H | F | H | CH | 0.209 (n = 1) | 12.9 |
| 1191–126 |  | H | CN | F | H | CH | 3.76 (n = 1) | >40 |

**Additional compounds:**

| Compound | Structure | EC$_{50}$ (µM) | CC$_{50}$ (µM) |
| --- | --- | --- | --- |
| 21-RS-17 |  | 1.73 ± 1.68 (n = 2) | >4 |
| R21 |  | >40 (n = 6) | >40 |

The online version of this article includes the following source data for Table 1:

**Source data 1.** JK series compound structures.

(prefix 'RS') was tested using in vitro *dye* release assays (*StGelais et al., 2007*) using genotype 1b p7 (J4 strain) (*Figure 1—figure supplement 4*). This confirmed that variation of the prototypic scaffold generated compounds displaying activity versus p7 channel function and that a specific structure-activity relationship (SAR) should be achievable. Cell culture assays confirmed compound activity and comprised the screening method for ensuing compound iterations (*Table 1*).

Finally, we compared JK3/32 with an amiloride derivative that has been progressed into early phase human trials in Asia. BIT225 was identified as an inhibitor of genotype 1a p7 using a bacterial screen and has been reported to display activity versus bovine viral diarrhoea virus (BVDV) (*Luscombe et al., 2010*), and more recently HCV in cell culture (*Meredith et al., 2013*). However, in our hands BIT225 showed no antiviral activity discernible from effects upon cellular viability (*Figure 1—figure supplement 5*); notably, no assessment of cellular toxicity was undertaken during previously reported HCV studies (*Meredith et al., 2013*), which used a concentration higher (30 µM) than the observed Huh7 CC$_{50}$ herein (18.6 µM) during short timescale assays (6–24 hr).

## JK3/32 SAR corroborates predicted binding to hairpin-based p7 channel models

We developed a library of JK3/32 analogues to explore SAR for inhibition of J4/JFH-1 secretion (*Table 1*). Of 41 compounds tested, twenty contributed directly to the JK3/32 SAR, which was largely consistent with energetically preferred in silico docking predictions (using Glide, Schrodinger). JK3/32 is predicted to bind into a predominantly hydrophobic cleft created between helices on the membrane-exposed site (*Figure 2a,b*). Predicted polar interactions occur between the side-chains of Tyr45 and Trp48 side and the carbonyl oxygen atom at the indole core (*Figure 2c*). Other predicted close contacts included residues experimentally defined by NMR to interact with rimantadine (*Foster et al., 2014*): Leu20, Tyr45, Gly46, Trp48, Leu50 and Leu52, and additional interactions with Ala11, Trp32 and Tyr42. Importantly, the majority of residues within this binding site are highly conserved; all residues are >90% conserved with the exception of Leu20 (45.67%) and Tyr45 (84.67%) (*Figure 2d*, *Figure 2—figure supplement 1*). However, unlike rimantadine, Leu20-Phe does not mediate resistance to this chemical series (*Foster et al., 2014*).

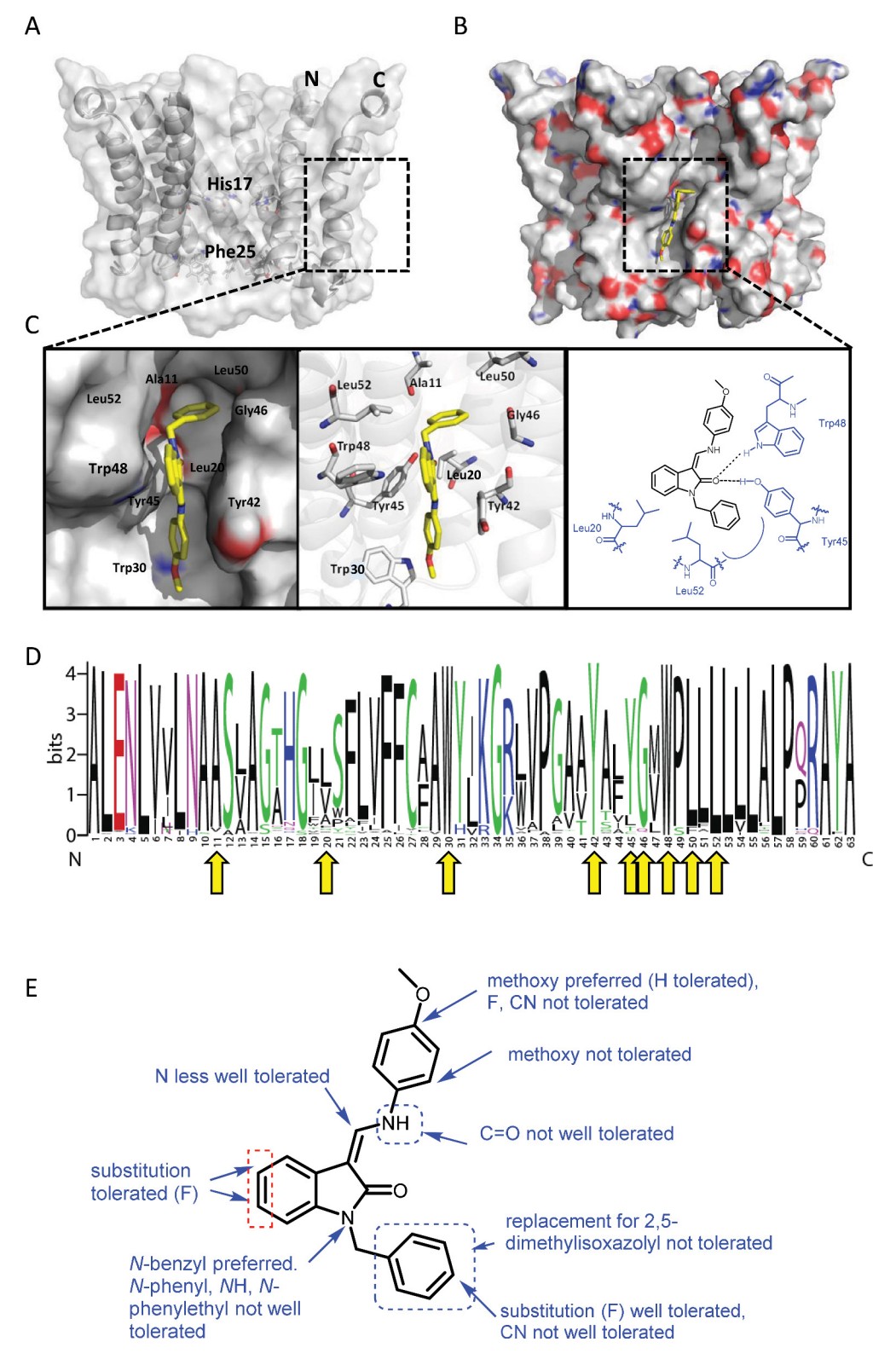

**Figure 2.** Predicted interactions of JK3/32 with genotype 1b p7 heptamer complexes. (A) Cutaway image of PDB: 3ZD0-based heptamer showing orientation of N- and C-terminal helices, predicted gating residue (Phe25) and proton sensor (His17). Box shows approximate region corresponding to peripheral drug-binding site. (B) Space-filling model of PDB: 3ZD0-based heptamer showing basic (blue) and acidic (red) charge distribution and positioning of JK3/32 (yellow) within peripheral binding site (box). (C) Zoomed images showing peripheral drug-binding site and predicted energetically

*Figure 2 continued on next page*

*Figure 2 continued*

preferred binding pose (in Glide) for JK3/32 (yellow) within membrane-exposed binding site as space fill (left), side chains (middle) and key interactions, including with Tyr45 and Trp48 (right). (D) Amino acid conservation within p7 across ~1500 sequences from the EU HCV database. Height corresponds to relative conservation of individual residues (quantified in *Table 1*). Yellow arrows indicate residues predicted to form direct interactions with JK3/32. (E) Summary of the structure-activity relationships for the JK3/32 series of inhibitors (*Table 1*) for their effects upon chimaeric GT1b HCV (J4/JFH-1) secreted infectivity following electroporation.

The online version of this article includes the following source data and figure supplement(s) for figure 2:

**Source data 1.** Jalview analysis data for EU HCV database p7 sequences.
**Figure supplement 1.** Graphical representation of percentage conservation for consensus p7 sequence (see *Figure 2—figure supplement 1—source data 1* for numerical summary).
**Figure supplement 1—source data 1.** Raw data for p7 sequence alignments.

JK3/32 SAR was consistent with its predicted binding pose (*Figure 2e* and *Table 1*) following docking within the peripheral binding site, defining key determinants of its activity. For example, substitution of the N1 position of the oxindole core demonstrated a preference for benzyl substitution (e.g. JK3/32) consistent with the group occupying a relatively large hydrophobic pocket between helices created by Leu and Ala residues. Incorporation of a longer, more hydrophobic group (N-ethylphenyl, 1191–137), was less well tolerated. Introduction of a NH (JK3-38), N-Ph (JK3-42) and a N-heterocyclic substituent (2,5-dimethylisoxazolylmethyl, 1191–106) abrogated antiviral effects. Attempts at substitution at the 'northern' phenyl ring was not well tolerated, with 4-OMe (e.g. JK3/32) or H (21-RS-8) preferred over 4-cyano (1191–112) and 2-methoxy (1191–104). 4-alkynyloxy was only moderately less active than 4-OMe (compare 1191–146 ($EC_{50}$ 2.48 μM) to 1191–140 ($EC_{50}$ 0.46 μM)), suggesting that further synthetic expansion from this site was possible, consistent with the modelling which directed this vector outwards from the binding pocket into the membrane. The linker at the 3-position of the oxindole core was sensitive to modification. The hydrazone analogue (21-RS-7) was much less active whilst replacement of the NH for a carbonyl group (21-RS-17) was also not well tolerated. This is consistent with the enamine linker adopting an important bridging unit for correct placement of the N1 substituent into the deep hydrophobic pocket. Substitution of the oxindole core at the 5- and 6-positions with F atoms (1191–124 and 1191–121, respectively) was generally well tolerated. Consistency between observed SAR and the heptameric 3ZD0-based model supports that rational design based upon this system generates authentic, specific p7-targeted antivirals; JK3/32 SAR was not consistent with the 2M6X structure (data not shown).

## Molecular dynamics supports stable JK3/32 binding at the peripheral site

We next undertook atomistic molecular dynamics simulations of genotype 1b p7 channel complexes with JK3/32 bound at the peripheral site to assess the stability of interactions predicted by docking studies over time. Encouragingly, atomistic simulations (100 ns) of JK3/32 bound to the peripheral site revealed marked stability of its binding pose, despite significant structural dynamics of the p7 bundle observed in hydrated lipid bilayers (*Figure 3a*). The root mean square deviation (RMSD) of the protein backbone Cα atoms indicated that the structural dynamics of JK3/32-bound p7 fluctuated within tolerable values, and were effectively indistinguishable from unbound protein. JK3/32 remained within the binding pocket throughout the course of the simulation, with the carbonyl group initially forming H-bonds with Tyr45 followed by subsequent bifurcation (after ~50 ns) with Trp48 (*Figure 3b*). The JK3/32 amino group made further interactions with various side chains over the course of the simulation.

Leu20Phe mutant p7 complexes also formed a stable interaction with JK3/32 (*Figure 3—figure supplement 1*), although Phe20 caused reorganisation and crowding of the binding pocket stabilised by π-π stacking interactions between Phe20 and Tyr45. The intramolecular H-bond between the NH and carbonyl oxygen was lost on simulation, although bifurcated H-bonding to Tyr45 and Trp48 was maintained. We infer that efficient H-bond formation is contributory to JK3/32 potency. By contrast, p7 with the Leu20Phe mutation disrupted binding of rimantadine within the pocket, with the drug failing to make H-bond contacts and leaving the pocket over the course of the simulation (*Figure 3—figure supplement 2*).

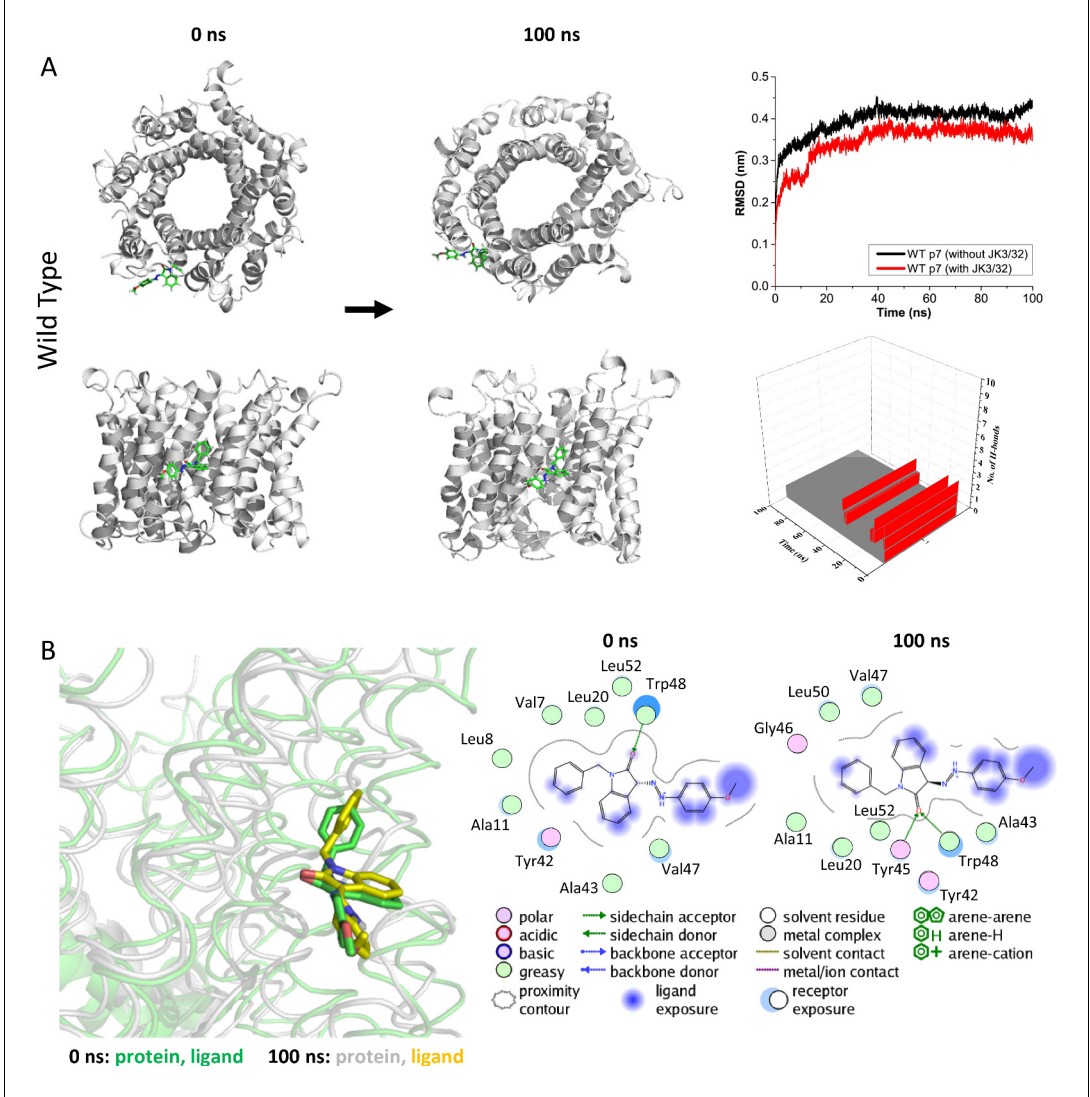

**Figure 3.** Atomistic simulations of JK3/32 interaction with PDB: 3ZD0-based heptamers. (**A**) 100 ns atomistic molecular dynamics simulation of p7 complexes bound to JK3/32, starting from a minimised pose in a hydrated lipid bilayer environment. Top-down and side views of complexes at the start and end of simulation are shown with a single molecule of JK3/32 bound at the membrane-exposed binding site. Graphs on the right show the root mean-squared variation (RMSD) over time (top) in the presence (red) or absence (black) of ligand, and the number of H-bonds formed throughout the simulation (bottom). (**B**) Overlay of p7 protein structures and JK3/32 orientation at 0 ns and 100 ns (left) alongside molecular interactions at the two time points (right); note bifurcation of H-bond interaction between indole carboxyl and Trp48 (0 ns) to both Trp48 and Tyr45 (100 ns).

The online version of this article includes the following source data and figure supplement(s) for figure 3:

**Source data 1.** Output data for MD simulations for Jk3/32 with wild type/L20F p7.

**Figure supplement 1.** Atomistic simulations of JK3/32 interaction with PDB: 3ZD0-based heptamers containing Leu20Phe polymorphism.

**Figure supplement 2.** Molecular Dynamic Simulation of rimantadine with heptameric 3ZD0-derived p7 and resistant Leu20Phe complexes.

**Figure supplement 2—source data 1.** Output data for rimantadine MD simulation.

## HCV entry is dependent upon p7 ion channel function

We previously demonstrated a link between p7 sequence and the acid stability of secreted HCV particles (*Atkins et al., 2014*). This prompted us to speculate that, in addition to its role during secretion, virion-resident channel complexes might influence virus entry. In agreement, JK3/32 reduced infectivity when added to infectious genotype 1b innoculae, but a structurally related analogue lacking activity versus genotype 1b or 3a, compound R21, did not (*Figure 4a,b*). However, inhibition of HCV entry occurred at higher JK3/32 concentrations compared to those effective

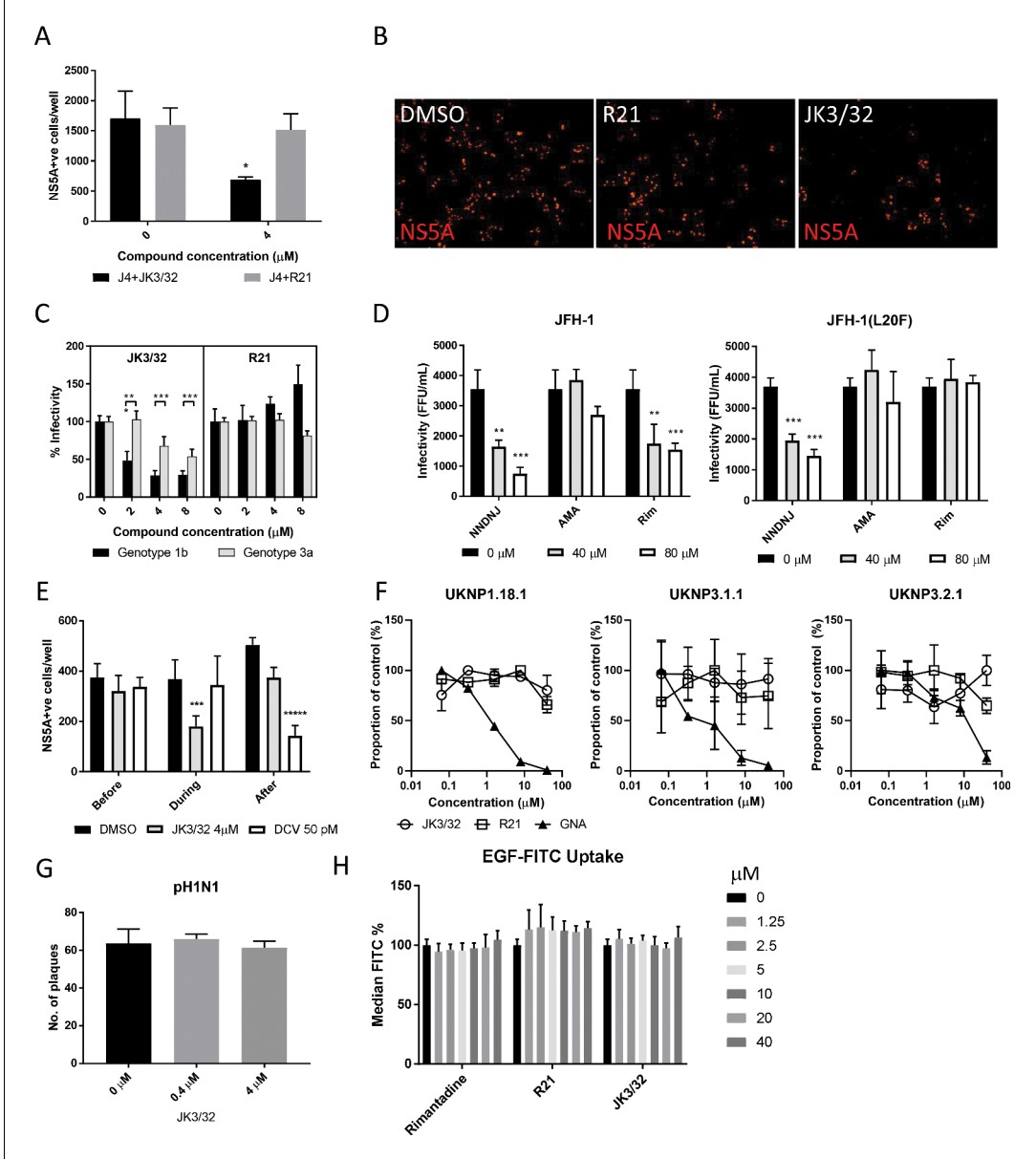

**Figure 4.** Characterisation of JK3/32 effects upon HCV entry. (**A**) Infectivity following application of JK3/32, or the inactive R21 analogue, during entry of GT1b chimaeric HCV into Huh7 cells. Virus innoculae were pre-treated with either compounds or DMSO for 20 min at room temperature prior to application to Huh7 cells overnight. Cells were washed extensively and assessed for infectivity 48 hr post infection by NS5A immunofluorescence, quantified using the Incucyte Zoom. (*$p \leq 0.05$, Student's t-test, n = 2). (**B**) Representative images from Incucyte analysis of NS5A-stained Huh7 cells in A for comparison. (**C**) Titrated JK3/32 concentrations were assessed during entry as described in A, comparing chimaeric GT1b (J4/JFH-1) and GT3a (S52/JFH-1) viruses (***$p \leq 0.001$, Student's t-test, n = 2). (**D**) Effects of prototypic p7 channel blockers against wild type and rimantadine-resistant GT2a HCV (JFH-1) during entry into Huh7 cells (**$p \leq 0.01$, ***$p \leq 0.001$, Student's t-test, n = 2). (**E**) JK3/32 effects upon entry when added prior (2 hr), during (4 hr) or post-infection (48 hr) with chimaeric GT1b HCV (J4/JFH-1), compared to Daclatasvir NS5Ai (***$p \leq 0.001$, *****$p \leq 0.00001$, Student's t-test, n = 2). (**F**) Lack of effect of JK3/32 or R21 upon entry of Lentiviral HCV pseudotypes into Huh7 cells compared to GNA positive control. HCV envelopes were derived from genotype 1b (UKNP1.18.1) or 3a (UKNP3.1.1 and UKNP3.2.1) patients. Results are representative of two experiments. (**G**) Lack of effect for JK3/32 upon IAV entry into MDCK cells. (**H**) Lack of effect for JK3/32, R21 or rimantadine upon clathrin-mediated endocytosis, measured by fluorescent EGF uptake.

The online version of this article includes the following source data and figure supplement(s) for figure 4:

**Source data 1.** Raw data for virus entry experiments.

**Figure supplement 1.** Additional control experiments for JK3/32 effects during HCV entry.

**Figure supplement 1—source data 1.** Raw data for virus entry control experiments.

against virion secretion making it necessary to confirm effects were both HCV- as well as p7-specific. Reassuringly, the same genotype-dependent variation in JK3/32 potency was evident during entry experiments as observed during secretion; genotype 3a chimaeric viruses required 2–4 fold higher concentration than genotype 1b and R21 again had no effect (*Figure 4c*). As expected, compounds displayed no evidence of cytotoxicity at higher concentrations (*Figure 4—figure supplement 1a,b*).

Multiple unsuccessful attempts were made to select resistance to JK3/32 in culture (data not shown), meaning that it was not possible to provide genetic evidence supporting interactions with this compound during virus entry. We infer that this is due to the high degree of conservation seen within the JK3/32 binding site (*Figure 2d*). However, we previously defined both strain- and polymorphism-dependent resistance to prototypic adamantane and alkyl imino-sugar p7i (*Foster et al., 2011*), providing a means to assess p7 target engagement during entry. Secretion of genotype 2a JFH-1 HCV is innately amantadine-resistant, yet remains sensitive to rimantadine and the alkyl imino-sugar *NN*-DNJ. Virus entry experiments recapitulated this pattern, with the latter two compounds displaying dose-dependent efficacy when added to infectious innoculae (*Figure 4d*). Moreover, entry of JFH-1 Leu20Phe was resistant to both amantadine and rimantadine, whilst remaining *NN*-DNJ-sensitive; *NN*-DNJ disrupts p7 oligomerisation rather than binding peripherally (*Foster et al., 2011*). Hence, p7 sequence dictated p7i-mediated blockade of HCV entry, providing genetic evidence for virion-resident channels. Consistently, JK3/32 only blocked infection when introduced during virus infection, concordant with a direct effect upon virion-resident channels, whereas the NS5Ai Daclatasvir only interfered with HCV infection when added post-entry (*Figure 4e*).

Next, we used Lentiviral vectors pseudotyped with HCV glycoproteins to establish whether JK3/32 effects were p7-specific, or might instead interfere with receptor binding, particle integrity or membrane fusion. Encouragingly, neither JK3/32 nor R21 added to innoculae had effects upon the entry of pseudotypes possessing patient-derived E1/E2 representing genotype 1b or 3a (*Urbanowicz et al., 2016a*), whereas *Galanthus nivalis* Agglutinin (GNA) blocked entry in a dose-dependent fashion (*Figure 4f*). Pseudotypes based upon prototypic genotype 1a or 2a, as well as the vesicular stomatitis virus glycoprotein (VSV-G) control, were also unaffected by JK3/32 or R21 (*Figure 4—figure supplement 1c*).

Lastly, to discern whether JK3/32 affected clathrin-mediated endocytosis, we tested sensitivity of pandemic H1N1 IAV entry to the compound. Not only does IAV enter cells via this pathway, it also retains virion-associated M2 viroporin proton channels, thereby serving as an additional control for JK3/32 specificity. Accordingly, JK3/32 did not affect IAV entry (*Figure 4g*). In addition, JK3/32 did not interfere with clathrin-dependent uptake of fluorescent-tagged epidermal growth factor (EGF) (*Figure 4h*, *Figure 4—figure supplement 1d*). Taken together, evidence supports that JK3/32 inhibits HCV entry in a p7 sequence-, genotype-, and temporally- dependent fashion, with no appreciable effects upon cellular endocytic pathways.

## Genotype-dependent R21 sensitivity supports the functional requirement for p7 channel activity during HCV entry

Unlike genotypes 1b and 3a, the genotype 2a JFH-1 strain in fact displays modest sensitivity to the R21 compound during secretion (*Figure 5A*). Accordingly, the compound also blocked JFH-1 entry when added to innoculae (*Figure 5B*). Such genotype-dependent variation in sensitivity to this inhibitor series was also observed in previous studies (*Foster et al., 2014*). Thus, whilst the R21 compound could not serve as a negative control during JFH-1 entry experiments, the discrete sensitivity compared to resistant 1b/3a channels implied that polymorphisms within the JFH-1-binding site underpinned this phenotype. Such genetic evidence for a compound distinct to JK3/32, yet targeting the same binding site, would further support not only the relevance of the peripheral binding site, but also the functional requirement for p7 channel activity during HCV entry.

Whilst highly conserved (*Figure 2—figure supplement 1*), the peripheral binding site does show very limited variation at positions 45, 50 and 51, and this is increased in genotypes more genetically distinct from 1b including genotype 2a isolates such as JFH-1. Compared to the J4 sequence, JFH-1 contains Y45T, L50F and L51C, whereas the S52 sequence contains Y45T and L51A; sequence divergence in other regions of the protein is more common. We therefore examined previously derived structure homology models (*Foster et al., 2014*) of p7 channels from these three genotypes to examine how sequence changes might affect predicted interactions with the R21 compound (*Figure 5C*). Interestingly, the Y45T/L50F change in JFH-1 reduced exposure of the R21 molecule to

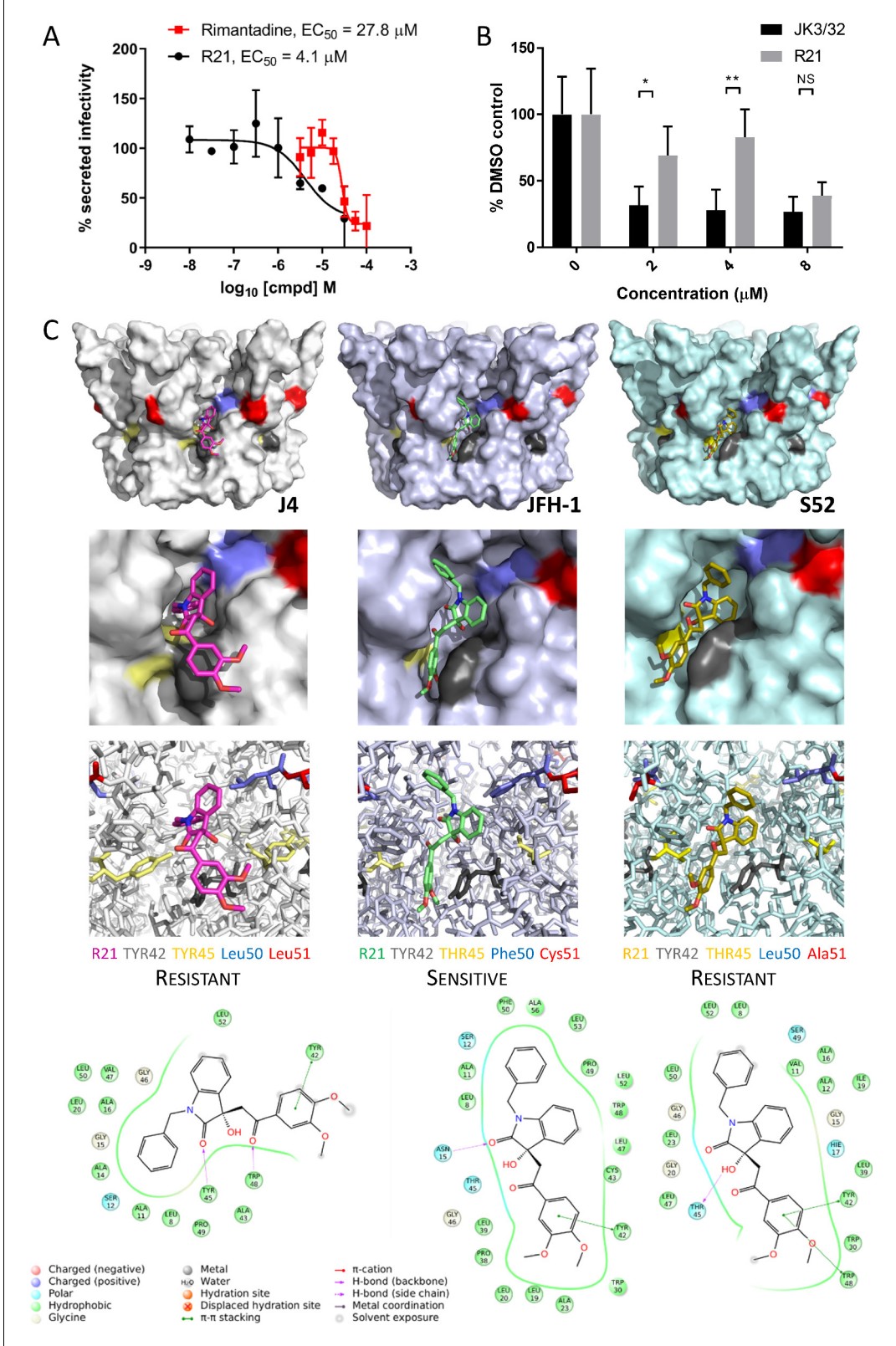

**Figure 5.** Susceptibility of genotype 2a JFH-1 HCV to R21 further supports a role during entry as well as peripheral site specificity. (**A**) Comparison of R21 potency vs. virion secretion of JFH-1 with rimantadine. Curves are representative of two experimental repeats, where each condition was carried out in quadruplicate and error bars represent standard deviations. (**B**) Infectivity following application of JK3/32 or R21 during JFH-1 entry into Huh7 cells. Virus innoculae were pre-treated with compounds or DMSO for 20 min at room temperature prior to application to Huh7 cells overnight. Cells

*Figure 5 continued on next page*

*Figure 5 continued*

were washed extensively and assessed for infectivity 48 hr post infection by NS5A immunofluorescence, then quantified using the Incucyte Zoom. (*p≤0.05, **p≤0.01 Student's t-test, n = 2). (C) Predicted energetically preferred binding poses and residue interactions for R21 within the peripheral binding pocket for genotype 1b (J4), 2a (JFH-1) and 3a (S52) HCV p7 structure models. Key residues are colour coded as shown, including positions 45 (yellow) and 50 (blue), where variants show a strong predicted influence upon R21 binding.

The online version of this article includes the following source data for figure 5:

**Source data 1.** Raw data for R21 docking interactions.

the surrounding environment, permitting a more stable interaction compared to the 1b or 3a channels; L51C was too distant from R21 to exert any obvious changes in binding. Furthermore, the Y45T change disrupts interactions between Y45 and Y42 seen in the 1b channel model allowing the dimethoxyphenyl group to better fit within the cavity. In addition, L50F prevents distortion of the benzyl group seen within the 1b structure, again promoting stability within the JFH-1-binding site. As genotype 3a lacks the L50F change, we infer that the Y45T/L50F dual polymorphism underpins JFH-1 R21 susceptibility.

We next speculated that the Y45T/L50F polymorphisms might simultaneously confer partial JK3/32 resistance as well as R21 sensitivity to genotype 1b channels. Given that we had been unable to select JK3/32 resistance in culture, the limited change in $EC_{50}$ between genotypes, and mutations in this region (e.g. Y42F, W48F) of the protein reduce particle production and infectivity (*Steinmann et al., 2007a*), we again used molecular modelling to investigate potential phenotypes. New structure homology models were generated for the genotype 1b (J4) channel complex, incorporating Y45T/L50F or the Y45T/L51C polymorphisms (*Figure 6*). Consistent with experimental data (*Figure 1C*), both of the dual polymorphisms led to reduced JK3/32- binding efficiency when present in the J4 background (*Figure 6A*), making it likely that Y45T alone might be responsible for this phenotype. By contrast, neither of the changes were able to enhance the binding of R21 to the genotype 1b channel complex (*Figure 6B*). We conclude, therefore, either that all three changes may be relevant despite the distance of position 51 from the binding site, or that overall structural differences within the JFH-1 channel are the correct context in which Y45T or the other polymorphisms might act. Note, only intra-, rather than inter-ligand binding differences are relevant due to the differential exposure of ligands to the environment, and distinct start-point binding poses within wild type channels.

Thus, along with rimantadine, these data support that three chemically distinct p7i undergo not only genotype-, but also polymorphism-specific interactions with the peripheral p7 binding site. This lends support to both the specificity of this inhibitor series, as well as a functional requirement for channel activity during virus entry.

## JK3/32 and a labelled derivative exert irreversible blockade of virion-resident p7 channels

Despite multiple experiments supporting a specific effect upon HCV entry, the elevated concentrations of JK3/32 required to block infection made it desirable to minimise cellular exposure to the compound and so comprehensively rule out off-target effects. Fortunately, our in-depth understanding of JK3/32 SAR, docking, and MD simulations indicated that the creation of a labelled tool compound should be feasible. We predicted that the northern 4-OMe group should tolerate the addition of a flexible linker without significant loss to binding affinity. Click chemistry was used to attach an azide reactive fluorophore (Alexa-Fluor 488 nm) to an alkynyl group at the 4-position of the phenyl ring generating a chemical probe (JK3/32-488), allowing compound concentrations to be calculated by fluorimetry.

Purified genotype 1b chimaeric HCV, dosed with 10 µM JK3/32, JK3/32-488, R21 or DMSO, was separated from unbound ligand by centrifugation through continuous iodixinol density gradients. Both control and R21-exposed gradients yielded overlapping peaks of infectivity at densities between 1.1 and 1.15 g/mL (corresponding to fractions 4–7 of the gradient), consistent with previous studies (*Figure 7a,b*, *Figure 7—figure supplement 1*). However, infectivity was effectively abrogated in peak fractions treated with either JK3/32 or JK3/32-488, although a small number of infected cells were detectable (*Figure 7a,b*, *Figure 7—figure supplement 1*). Quantification of the fluorescent signal from JK3/32-488 revealed concentrations within peak infectivity fractions were <2

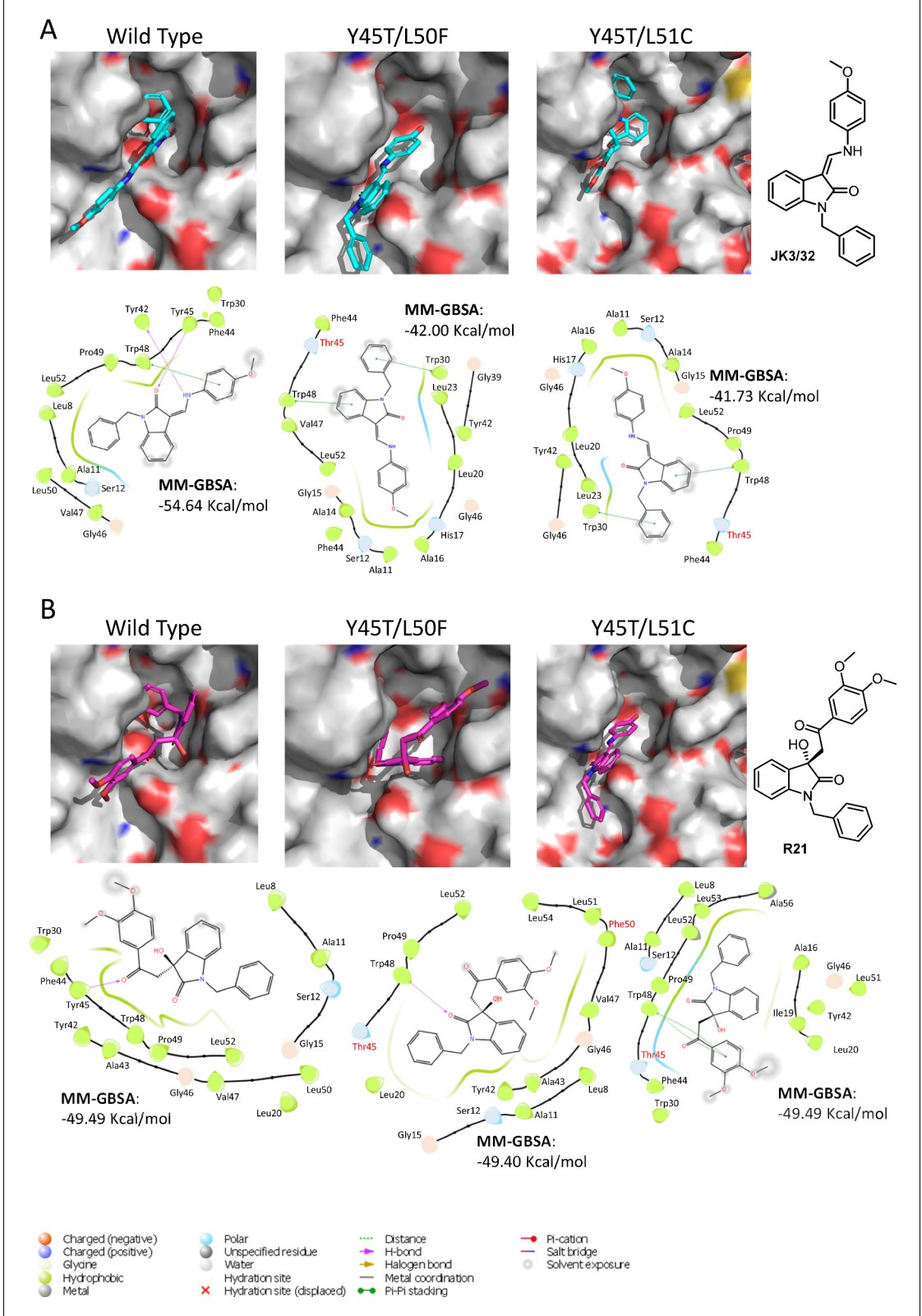

**Figure 6.** Predicted binding of JK3/32 and R21 to mutated J4 p7 channel complexes. Structure homology models of the genotype 1b J4 p7 channel complex containing either Y45T/L50F or Y45T/L51C double mutations, were generated and energy-minimised using Maestro (Schrödinger). (**A**) JK3/32 was re-docked into the peripheral binding site for the wild type or mutated channel complexes using Glide (see Materials and methods), identifying interactions within the binding site. Predicted binding energies were calculated using MM-GBSA. (**B**) As for A, but for the R21 ligand.

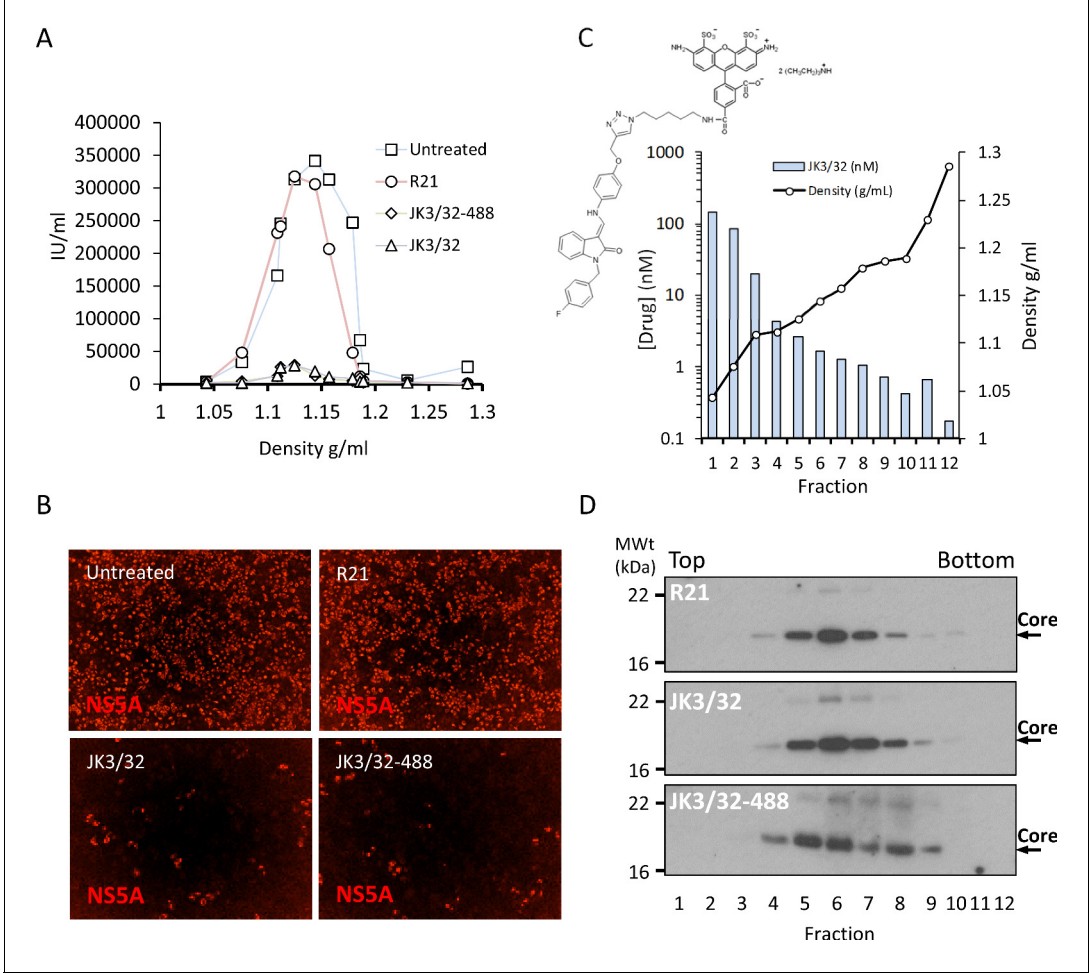

**Figure 7.** Direct JK3/32 virion dosing reduces HCV infectivity with minimal exposure of compound to cells. Concentrated, purified high titre chimaeric GT1b HCV (J4/JFH-1) was incubated with DMSO, active JK3/32 (-488) or control compounds (R21) at 10 μM for 20 min prior to separation on a continuous 10–40% iodixinol/PBS density gradient followed by fractionation. (**A**) IncuCyte quantitation of NS5A immunofluorescence staining within each fraction expressed as infectious units (IU) per mL. (**B**) Immunofluorescence staining of HCV NS5A protein within naïve Huh7 cells 48 hr post-infection with 10 μL of fraction 6, corresponding to peak infectivity for untreated control gradients. (**C**) Calculated concentration of JK3/32-488 (chemical structure shown in inset) based upon fluorimetry within each of 12 fractions taken from the top of the gradient and corresponding fraction densities. (**D**) Western blot analysis of drug treated gradient fractions, probing for HCV core protein.

The online version of this article includes the following source data and figure supplement(s) for figure 7:

**Figure supplement 1.** IncuCyte image set from iodixinol gradients shown in *Figure 7*.

**Figure supplement 1—source data 1.** Raw image data for gradient immunofluorescence.

nM (fractions 5 and 6). Hence, separation of treated virions from unbound compound using this method resulted in cellular exposure to far lower concentrations than those able to block HCV secretion, making off-target effects extremely unlikely (*Figure 7c*). Importantly, neither JK3/32 nor JK3/32-488 affected the stability of HCV particles as migration of virion-associated core protein corresponded with the peak of infectivity in each case, and was identical to the R21 control (*Figure 7d*). Hence, we conclude that effects upon HCV entry are not attributable to cellular exposure to JK3/32, and that the compound irreversibly blocks virion-resident p7 channels.

## Discussion

Our rationally derived lead, JK3/32, and the extensive chemical SAR series that underpins it, represent a dramatic leap forward compared with previously described p7i on several fronts. First, correlation between SAR and channel models constructed using hairpin-shaped protomers (based upon the

3ZD0 monomeric NMR structure), rather than the 2M6X hexamer, supports their biological relevance. Second, the SAR across the series provides evidence for a number of structurally related analogues with defined potency and genotype specificity; whilst previously reported initial hit compounds showed similar potency in some cases, their structural diversity meant that they lacked a common scaffold which made it harder to identify a consensus pharmacophore for binding (*Foster et al., 2014*). Third, our understanding of SAR and the way in which JK3/32 is predicted to interact with the p7 channel complex, enabled creation of the first – to our knowledge – labelled tool compound based upon a viroporin inhibitor. The success of this molecule further corroborated our understanding of SAR and predicted interactions between p7i and structure models of the p7 channel complex. Lastly, drugs based upon JK3/32 would represent antivirals with a distinct mechanism of action compared with current HCV DAA that target the viral replicase. Moreover, these may potentially target two discrete stages of the virus life cycle, namely virion secretion and entry. Hence, this work sets a new precedent for viroporin-targeted drug development that should enable the field to advance beyond the previous clinical precedent set by amantadine- and/or rimantadine-based influenza therapy.

The potency and selectivity demonstrated by JK3/32 was consistent with the SAR demonstrated within the series, docking predictions within the peripheral binding site, as well as MD simulations of the bound JK3/32 ligand over time. Each binding site occurs between adjacent hairpin-formation protomers and the majority of residues are highly conserved (*Figure 2c,d*). This includes Tyr45 and Trp48, which form key H-bond interactions with the JK3/32 indole carbonyl. Only Leu20 (~46% conserved) shows appreciable variation within the binding site, with the rimantadine-resistant Phe20 occurring in just ~3.6% of isolates in the EU HCV database (*Figure 2d*, *Figure 2—figure supplement 1*); Val (~30%) or Ala (~12%) are more common. Moreover, Phe20 stabilises rather than disrupts interactions with JK3/32 during MD simulations, and JK3/32-related compounds are unaffected by this mutation in culture (*Foster et al., 2014*). We infer that the high degree of conservation within this site explains why JK3/32 specific resistance was not selectable in culture, despite several attempts.

In addition to L20F adamantane resistance, HCV also displays genotype-dependent resistance to other inhibitor classes; for example, genotype 3a resistance to alkyl imino-sugars helped to identify a F25A resistance polymorphism (*Foster et al., 2011*). By contrast, the sensitivity of JFH-1 p7 to rimantadine, but not amantadine, is not easily attributable to a single residue, suggesting that multiple interactions and/or the global structure of the channel complex may play a role (*Griffin et al., 2008*; *Foster et al., 2011*; *StGelais et al., 2009*). Whilst relative JK3/32 resistance occurred for genotypes 2a (JFH-1) and 3a (S52) compared to 1b (*Figure 1c*), JFH-1 p7 alone displayed discrete sensitivity to R21 (*Figure 5a,b*). Comparison of amino acid variability within the peripheral site highlighted Y45T/L50F as a dual polymorphism potentially capable of influencing R21 binding (*Figure 5c*). Interestingly, introducing these changes into the genotype 1b channel model led to reduced JK3/32 binding, consistent with experimental data, yet did not increase predicted binding of R21 to the J4 channel (*Figures 1c* and *6*). Thus, whilst engineering such mutations into the J4 p7 background may lead to a modest increase in JK3/32 resistance as seen for JFH-1, a discrete resistance phenotype may be more complex to achieve and may incur a significant fitness cost as seen for mutations proximal to this region (*Steinmann et al., 2007a*). A requirement for two or more changes may also explain the high genetic barrier in JK3/32 resistance experiments. Nevertheless, the discrete resistance phenotype for the R21 compound supports the relevance of the peripheral binding site and provides orthogonal validation that p7 channel activity is required during HCV entry. In addition, should resistance emerge in the future to JK3/32 progeny involving positions 45 and 50, this could conceivably be countered using compounds evolved from the R21 chemotype alongside in combination therapies. This further demonstrates the added value of the JK3/32 series SAR compared with other prototypic inhibitors.

It was interesting to note that JK3/32 SAR was compatible with 3ZD0-based hairpin-based channel models, but not those based upon the genotype 5a (EUH1480 isolate) 2M6X channel complex solution structure. 2M6X comprises triple-helix protomers and also contains Phe20 within a peripheral adamantane binding site (*OuYang et al., 2013*), although this site differs significantly in amino acid composition and structure compared to the 3ZD0-based hairpin heptamer model. Unfortunately, the recombinant protein used for structure determination did not display ion channel activity, making it impossible to determine drug sensitivity (*OuYang et al., 2013*). Nevertheless, a related

genotype 5a (SA-13) also retains Phe20 and is rimantadine-sensitive in culture (*Meredith et al., 2013*), supporting that Phe20 mediated rimantadine resistance is context-specific. The EUH1480 p7 sequence was modified to enhance recombinant expression, primarily at Cys residues. Amongst these, Cys2 is only found within GT5a (present in just 0.07% of p7 sequences overall, see *Figure 2— figure supplement 1*), and Cys27 is highly conserved across all genotypes (>95%). It is conceivable that these mutations were responsible for the lack of observable channel activity for this protein.

Unlike the peripheral rimantadine binding site in 3ZD0-based models, the region bound by amantadine in the 2M6X complex (*OuYang et al., 2013*), comprising Phe20, Val25, Val26, Leu52, Val53, Leu55 and Leu56, is poorly conserved (*Figure 2—figure supplement 1*). Only Leu52 and Leu55 represent consensus residues at their respective positions, with all the others being minority species. Moreover, three of the seven residues vary between the otherwise closely related SA-13 and EUH1480 genotype 5a strains (positions 26, 53, and 55). Unsurprisingly, JK3/32 binding and SAR is incompatible with this site within the 2M6X complex, or more importantly, with a genotype 1b homology model based upon the 2M6X structure (data not shown). Taken together, our findings add to the NMR-specific concerns recently raised regarding 2M6X (*Oestringer et al., 2018*) and support that biologically functional p7 channels comprise arrangements of hairpin protomers.

JK3/32 was compared to a patented p7i advanced into early phase human studies in South East Asia, BIT225 (*Luscombe et al., 2010*) (N-Carbamimidoyl-5-(1-methyl-1H-pyrazol-4-yl)−2-naphthamide, Biotron Ltd, Australia). BIT225 is an amiloride-related compound selected as an inhibitor of genotype 1a p7 (H77 strain) from a bacterial screen, with activity against recombinant protein at 100 µM in suspended bilayers. As a surrogate for HCV in culture, BIT225 potently inhibited replication of the *Pestivirus*, BVDV ($EC_{50}$314 nM), although specificity was not attributed to the BVDV p7 protein (*Luscombe et al., 2010*). However, BIT225 later showed genotype-dependent efficacy versus HCV (*Meredith et al., 2013*), with $EC_{50}$ values of 10 µM or 30 µM (genotype 2a and 5a chimaeric HCV, respectively); interestingly, these concentrations are far higher than those effective versus BVDV (*Luscombe et al., 2010*). Whilst we observed a similar anti-HCV $EC_{50}$ of 17.7 µM versus genotype 1b chimaeric HCV, this coincided with the $CC_{50}$ of 18.6 µM (*Figure 1—figure supplement 5*), making it impossible to define a selective antiviral effect. Notably, Huh7 cell cytotoxicity was not determined in previous HCV studies (*Meredith et al., 2013*), whereas BVDV studies reported a $CC_{50}$ of 11.6 µM for Madin-Darby bovine kidney (MDBK) cells, with appreciable (~16%) cytotoxicity at just 4 µM (*Luscombe et al., 2010*). We conclude that BIT225 may show greater potency versus BVDV than HCV, and that selective antiviral effects cannot be determined for HCV in the Huh7 system. This contrasts with the considerable selectivity index observed for JK3/32 (>500, *Figure 1—figure supplement 2*).

JK3/32 inhibition of secreted HCV infectivity showed cross-genotypic potency, including versus genotype 3a, which can be more difficult to treat using current DAAs (*Figure 1b,c*). Thus, we were naturally cautious regarding the higher concentrations required to block HCV entry, although this effect was reproducible compared to the inactive R21 analogue for genotypes 1b and 3a, showed genotype-dependent variation and was coincident with entry (*Figure 4a–c,e*). Lentiviral pseudotype assays supported that JK3/32 did not affect E1/E2 mediated receptor binding or membrane fusion (*Figure 4f*, *Figure 4—figure supplement 1c*), plus it did not affect IAV entry or clathrin-mediated endocytosis (*Figure 4g,h*). Moreover, strain/polymorphism-dependent resistance to prototypic p7i and the R21 compound provided genetic evidence of target engagement (*Figures 4d* and *5b*) and we were effectively able to rule out cell-dependent effects using our fluorescent-tagged JK3/32 derivative (*Figure 7*), supporting irreversible dosing of virion-resident p7 channels.

We speculate that the different biological roles of p7 channel activity during secretion and entry may explain why higher p7i concentrations are required to block the latter process. During secretion, p7 must counteract active proton gradients generated by vATPase in order to prevent acidification within large secretory vesicles (*Wozniak et al., 2010*), conceivably requiring a considerable number of viroporin complexes. By contrast, during entry, virion-resident p7 would promote core acidification in the same direction as proton gradients generated within endosomes, presumably achieved by a small number of channel complexes. Thus, inhibiting a fraction of p7 channels within a secretory vesicle might prevent vATPase antagonism, whereas saturating inhibitor concentrations would be required during entry to prevent activation of even a single virion-resident p7 channel. This hypothesis may also explain why cell-to-cell infection appears less sensitive to p7i as virions presumably bypass acidifying secretory vesicles (*Meredith et al., 2013*).

Studies of HCV harbouring mutations within the highly conserved p7 basic loop (consensus: K33, R35, p7 sequence) have provided evidence contrary to the presence of p7 channels within virions (*Steinmann et al., 2007a*). Specifically, highly efficient chimaeric J6/JFH-1 HCV harbouring basic loop mutations is able to produce secreted infectious virions, albeit at significantly reduced titre (~4-log10 reduction). Basic loop mutant particles displayed equivalent specific infectivity compared with wild type, suggesting that p7 function is not required during entry. Moreover, in the less efficient JFH-1 background where basic loop mutants effectively abrogate secreted infectivity, restoration of a limited level of secreted infectivity occurred upon either *trans*-complementation with IAV M2, or Bafilomycin A (BafA) treatment of infected cells (*Bentham et al., 2013*; *Wozniak et al., 2010*). These observations suggest that supplementing p7 function during virion secretion is sufficient to restore infectivity, negating the requirement for p7 to act during entry.

However, this interpretation assumes that basic loop mutations specifically affect ion channel activity in isolation, yet no published evidence exists to support this notion. Instead, basic loop mutations induce defects in p7 stability and E2-p7-NS2 precursor processing within infected cells (*Bentham et al., 2013*), as well as severe disruption of membrane insertion observed in vitro (*StGelais et al., 2009*). It is likely that cell phenotypes directly link to defective membrane insertion, suggesting that p7 may spontaneously insert into membranes rather than depending upon the signal recognition particle. Spontaneous membrane insertion would explain why p7 lacking an upstream signal peptide forms functional channels within cells (*Griffin et al., 2004*), as well as how HCV harbouring deletions of half of the p7 protein remain viable for replication despite the predicted effect of such mutations upon polyprotein topology (*Brohm et al., 2009*). Thus, whilst only a minority of basic loop mutant p7 proteins might successfully insert into membranes, this should result in the formation of functional channel complexes. This scarcity of p7 complexes would profoundly disrupt virion secretion in the model proposed above due to an inability to counteract vATPase. However, a minority of virions might conceivably contain sufficient functional p7 complexes. Provision of M2 and/or BafA would therefore rescue secretion of such particles by allowing them to survive within acidifying vesicles, which could then proceed to infect naïve cells.

One limitation of our study is the absence of p7 immunological detection within purified HCV virions. We previously generated the only published p7-specific polyclonal antisera, which required extensive concentration to detect relatively high protein concentrations (*StGelais et al., 2009*). Nevertheless, these are the only reagents described for the detection of native p7 protein, as confirmed by others (*Jirasko et al., 2010*; *Luik et al., 2009*). Whilst we applied these reagents to virion detection, stocks expired prior to a conclusive outcome. Others used recombinant viruses expressing epitope tagged p7 to investigate the presence of channel complexes within virions by co-immunoprecipitation (*Vieyres et al., 2013*). However, whilst p7 was not detectable within either anti-ApoE or Anti-E2 precipitates, the levels of core protein present were also low; by analogy with M2, we expect p7 to be present in stoichiometrically low numbers compared with canonical virion components. Similarly, p7 was not detected by mass spectrometry (MS) conducted upon affinity-purified HCV particles (*Lussignol et al., 2016*), yet this was also true for the E1 glycoprotein, which should show equivalent abundance to E2 resulting from heterodimer formation.

Thus, poor reagent quality combined with low protein abundance hampers direct detection of p7 within virions, representing a significant challenge that we intend to address by the addition of an affinity tag to the 'click'-labelled JK3/32 derivative. Concentration of virions and signal amplification will be essential as gradient infectivity peaks have an approximate particle molarity of 0.5 fM, based upon infectivity measurements (*Figure 7a*). IAV particles contain only 16–20 M2 proteins (*Pinto and Lamb, 2006*), meaning that even a high particle: infectivity ratio of 1000:1 would likely yield a maximal virion-associated p7 concentration in the tens of pM. Hence, a virion-associated fluorescence signal was not observed within peak infectivity fractions where the baseline unbound JK3/32-488 concentration was ~2 nM.

The favourable potency and selectivity index demonstrated by JK3/32 could serve as a high quality starting point for a more comprehensive p7-targeted drug development programme. Blocking p7 activity would interfere with two distinct stages of the virus life cycle compared to existing replicase-targeted DAAs, namely virion secretion and entry. Hence, p7i could be ideal for delivering effective antiviral prophylaxis against de novo exposure (e.g. needle stick/iatrogenic or perinatal) or transplanted graft re-infection, in addition to use alongside conventional DAA combination therapies treating chronic infection. Whilst blocking both entry and secretion may require higher plasma

concentrations compared with targeting secretion alone, the favourable selectivity index for JK3/32 suggests this should be feasible and future development may further optimise potency and bioavailability.

In summary, viroporins represent an untapped reservoir of antiviral drug targets that has historically been under-explored following the shortcomings of adamantane M2 inhibitors. Our work shows that it is possible to take a step-change in targeting these proteins, providing a new approach to antiviral therapy and generating novel research tools with which to dissect viroporin function.

# Materials and methods

## Key resources table

| Reagent type (species) or resource | Designation | Source or reference | Identifiers | Additional information |
|---|---|---|---|---|
| Strain, strain background (include species and sex here) | Hepatitis C virus full length infectious cell culture clone: JFH-1 | *Wakita et al., 2005*. (via Apath) under MTA | GenBank: AB047639.1 | |
| Strain, strain background (include species and sex here) | Hepatitis C virus full length infectious (chimaeric) cell culture clone: J4/JFH-1 | PMID:21177811, direct from Prof Bukh under MTA | GenBank: JF343781.2 | |
| Strain, strain background (include species and sex here) | Hepatitis C virus full length infectious (chimaeric) cell culture clone: S52/JFH-1 | PMID:21177811, direct from Prof Bukh under MTA | GenBank: JF343784.2 | |
| Strain, strain background (include species and sex here) | Influenza A/England/ 195/2009 (E195) | Prof Wendy Barclay, Imperial College | NCBI:txid645582 | Virus stock |
| Genetic reagent (include species here) | Hepatitis C virus full length infectious (chimaeric) cell culture clone: JFH-1 (L20F) | PMID:21520195 | Modified GenBank: AB047639.1 | Adamantane resistant p7, JFH-1 genotype 2a HCV |
| Cell line (include species here) | Huh7 (human hepatocellular carcinoma) | JCRB, via Prof John McLauchlan, CVR Glasgow | JCRB0403 | Mycoplasma tested and STR profile in-house |
| Cell line (include species here) | Madin-Derby Canine Kidney (MDCK) cells | ATCC | ATCC CCL34 | As provided |
| Cell line (include species here) | Human Embryonic Kidney (HEK) 293 T cells | ATCC | ATCC CRL-3216 | Adenovirus transformed, expressing SV40 large T antigen (G418$^r$), as provided |
| Transfected construct (include species here) | (MLV) based Lentiviral luciferase reporter vector (pTG126) | DOI: 10.1128/JVI.02700-15 | | MLV Lentiviral reporter pseudotyped with HCV envelope glycoproteins. |
| Transfected construct (include species here) | MLV Gag-Pol expressing plasmid (phCMV-5349) | DOI: 10.1128/JVI.02700-15 | | MLV Lentiviral reporter pseudotyped with HCV envelope glycoproteins. |
| Transfected construct (include species here) | vesicular stomatitis virus glycoprotein (VSV-G) expressing plasmid | DOI: 10.1128/JVI.02700-15 | | MLV Lentiviral reporter pseudotyped with HCV envelope glycoproteins. |
| Transfected construct (include species here) | pcDNA3.1D-E1/E2 encoding HCV envelope from genotype 1a (H77) | DOI: 10.1128/JVI.02700-15 | | MLV Lentiviral reporter pseudotyped with HCV envelope glycoproteins. |

*Continued on next page*

*Continued*

| Reagent type (species) or resource | Designation | Source or reference | Identifiers | Additional information |
|---|---|---|---|---|
| Transfected construct (include species here) | pcDNA3.1D-E1/E2 encoding HCV envelope from genotype 2a (JFH-1) | DOI: 10.1128/JVI.02700-15 | | MLV Lentiviral reporter pseudotyped with HCV envelope glycoproteins. |
| Transfected construct (include species here) | pcDNA3.1D-E1/E2 encoding HCV envelope from genotype 2a (J6) | DOI: 10.1128/JVI.02700-15 | | MLV Lentiviral reporter pseudotyped with HCV envelope glycoproteins. |
| Transfected construct (include species here) | pcDNA3.1D-E1/E2 encoding HCV envelope from patient isolate UKNP1.18.1 (genotype 1b) | DOI: 10.1128/JVI.02700-15 | | MLV Lentiviral reporter pseudotyped with HCV envelope glycoproteins. |
| Transfected construct (include species here) | pcDNA3.1D-E1/E2 encoding HCV envelope from patient isolate UKNP3.2.1 (genotype 3a) | DOI: 10.1128/JVI.02700-15 | | MLV Lentiviral reporter pseudotyped with HCV envelope glycoproteins. |
| Transfected construct (include species here) | pcDNA3.1D-E1/E2 encoding HCV envelope from patient isolate UKNP3.1.1 (genotype 3a) | DOI: 10.1128/JVI.02700-15 | | MLV Lentiviral reporter pseudotyped with HCV envelope glycoproteins. |
| Antibody | (Include host species and clonality) Sheep anti-NS5A, polyclonal serum | DOI: 10.1074/jbc.M210900200 | | (include dilution) 1. diluted 1/2500 for immunofluorescence |
| Antibody | Monoclonal Mouse anti-HCV core, C7-50 | Thermo Fisher | catalogue # MA1-080 | Diluted 1/1000 for western blot |
| Antibody | AlexaFluor 594 nm Donkey anti-Sheep antibody | Thermo Fisher | catalogue # A-11016 | Diluted 1/500 |
| Antibody | HRP conjugated Goat anti Rabbit secondary | Thermo Fisher | cat # G-21234 | Diluted 1/5000 |
| Peptide, recombinant protein | Hepatitis C virus genotype 1b, J4 isolate, FLAG-p7 (N-terminal tag) | DOI: 10.1002/hep.26685 DOI: 10.1074/jbc.M602434200 | | In-house expression. Cleaved from GST, HPLC purified |
| Commercial assay or kit | T7 Ribomax In vitro transcription kit | Promega | cat # P1320 | |
| Commercial assay or kit | Firefly luciferase assay system | Promega | cat# E1500 | |
| Chemical compound, drug | Rimantadine hydrochloride | SIGMA | 1604508 | |
| Chemical compound, drug | Amantadine hydrochloride | SIGMA | A1260 | |
| Chemical compound, drug | NN-DNJ | Toronto Biochemicals | N650300 | |
| Chemical compound, drug | JK3/32 series | This paper. Synthesised in-house, available upon request, subject to MTA and stock availability | | |
| Chemical compound, drug | BIT225 | This paper. Synthesised in-house, available upon request, subject to MTA and stock availability | | |
| Software, algorithm | Maestro with the OPLS3 force field and solvation model VSGB | Schrödinger | RRID:SCR_016748 | |

*Continued on next page*

*Continued*

| Reagent type (species) or resource | Designation | Source or reference | Identifiers | Additional information |
|---|---|---|---|---|
| | Protein Preparation Wizard (Version 12.3.13) | Schrödinger | RRID:SCR_016748 | |
| | 'Ligand Docking' tool | Schrödinger | RRID:SCR_016748 | |
| | Prime MM-GBSA tool in Schrödinger Maestro | Schrödinger | RRID:SCR_016748 | |
| | MOE software (Version 2015:1001) | http://www.chemcomp.com | RRID:SCR_014882 | |
| | AMBER 94 force field | *J. Am. Chem. Soc.* **1995**, 117, 5179 | RRID:SCR_018497 | |
| | MMFF94 force field | | RRID:SCR_015986 | |
| | Amber ff99SB-ILDN force field (ff) | | RRID:SCR_018497 | |
| | Gromacs 4.5.5 | | RRID:SCR_014565 | |
| | Jalview | http://www.jalview.org | RRID:SCR_006459 | |

## Virus secretion inhibition assays

Huh7 cells were a kind gift from Professor John McLauchlan, MRC/University of Glasgow Centre for Virus Research (CVR), confirmed via in-house STR profiling. Cells were cultured and propagated as described previously (*Griffin et al., 2008*). Cells were checked regularly (every three months) for mycoplasma contamination using an in-house service. Secreted infectivity was measured as described (*Stewart et al., 2015*). Briefly, 1 µg of linearised HCV constructs pJFH1, pJFH-1(L20F), pJ4/JFH1, or pS52/JFH1 (*Foster et al., 2011*; *Gottwein et al., 2011*; *Wakita et al., 2005*) was used to perform in vitro transcription (RiboMax express, Promega) following the manufacturer's protocol. Following phenol/chloroform extraction, $4 \times 10^6$ Huh7 cells were electroporated with 10 µg HCV RNA. Electroporated cells were seeded at $2.5 \times 10^4$ cells/ well in 100 µL volume in 96 well plates and incubated 4 hr. Compound dose responses were prepared at 400x in DMSO, diluted 1:20 into media in an intermediate plate, and 1:20 into the final cell plate to yield final 0.25% (v/v) DMSO. All compound treatments were dosed in duplicate. Dosed cells were incubated 72 hr before performing 1:4 dilution (50 µL) of virus-containing supernatant onto a plate of naïve Huh7 cells (150 µL), seeded at $8 \times 10^3$ cells/well 6 hr previously. For cytotoxicity analysis, producer plates were washed in PBS and fixed in 4% PFA prior to imaging cellular confluency using an IncuCyte ZOOM (Essen BioSciences). Infected Huh7 cells were incubated 48 hr before washing 3x in PBS and fixing in 4% PFA. Fixed cells were washed in PBS and permeabilised using 0.1% Triton X-100 (v/v) in PBS for 10 min, RT. Following PBS wash, cells were immuno-stained for NS5A to quantify infected cells. Anti-NS5A antibody was used at 1:2000 in PBS supplemented with 10% FBS, 16 hr, 4 °C. Following 3x PBS washes, Alexa-Fluor 594 nm Donkey anti-Sheep antibody was added at 1:500, 2 hr, RT under subdued light. Cells were washed in PBS and imaged using phase and red channels (IncuCyte ZOOM). Infected cells positive for NS5A expression were quantified using IncuCyte parameters previously described (*Stewart et al., 2015*), normalised to DMSO control and non-linear regression fitted using Prism 6 (GraphPad) to determine $EC_{50}/CC_{50}$.

Each experiment included a serial dilution of untreated virus confirming 1:4 dilution fell within a linear range and internal DAA $EC_{50}$ controls (data not shown). Determined Z-factor was routinely >0.45. In addition, the Incucyte cell confluence tool was used as a measure of cell viability in both transfected/inhibitor-treated producer cells and target cell populations; producer cell plates were also subjected to MTT (3-(4,5-dimethylthiazol-2-yl)−2,5-diphenyltetrazolium bromide) assays to determine potential compound effects against Huh7 cell metabolism. Lastly, analogous plates were set up using Huh7 cells electroporated with a subgenomic HCV replicon (genotype 2a JFH-1 strain – the same NS3-5B replicase as present within chimaeric HCV) encoding firefly luciferase in the first open reading frame. 48 hr post-electroporation, cells were lysed and luciferase activity determined using a commercially available kit according to the manufacturer's instructions (Promega).

## HCV entry inhibition assays

HCV entry experiments used virus supernatant stocks harvested 72 hr post-electroporation and stored at −80 °C. Huh7 cells were seeded at $8 \times 10^3$ cells/ well in 100 μL in 96-well plates for 6 hr. Indicated compound was added to virus-containing supernatant stocks immediately prior to infection at MOI of 0.6 and incubated 18 hr. For 'Time of Addition' experiments (*Figure 4e*), inhibitors were added for 2 hr *before*, for 4 hr *during*, and for 48 hr *after* infection, as indicated. Cells were washed and incubated 48 hr in media prior to quantifying infected cells via NS5A immunostaining as described above. Statistical significance was determined for test samples vs. controls using unpaired, two-tail Student's t-tests. Note, incubation for 48 hr was necessary to achieve robust and reproducible numbers of infected cells for counting in IncuCyte assays; shorter assays led to unacceptable errors in quantitation. However, as shown in *Figure 1—figure supplements 1* and *2*, the assay was proven to discriminate p7-dependent effects from those resulting from impaired RNA replication or cell viability.

## Liposome dye release assay

Recombinant genotype 1b p7 (J4 strain) was expressed and purified as described previously (*Clarke et al., 2006*). Channel activity was assessed using liposome dye release assays (*Griffin et al., 2008*; *StGelais et al., 2007*; *StGelais et al., 2009*), mixing protein with liposomes containing a self-quenching concentration of carboxyfluorescein and monitoring ensuing gain in fluorescence as an indirect measure of p7 activity.

## Influenza A virus plaque reduction assay

Madin-Darby Canine Kidney (MDCK, from ATCC) cells (seeded at $5 \times 10^5$/ well of a 12 well plate 4 hr prior to infection) were infected for 1 hr with A/England/195/2009 (E195) influenza A virus (IAV) at a multiplicity of infection (MOI) of 0.01, following preincubation with compounds for 30 min on ice. Virus-containing media was then removed and replaced with serum free (SF) minimal essential media (MEM) with 1 μgmL$^{-1}$ TPCK trypsin containing compounds for 24 hr. Clarified supernatant dilutions of $10^{-1}$ to $10^{-4}$ were then used to infect fresh monolayers of MDCK for 1 hr, then replaced with a 3:7 mixture of 2% w/v agar (Oxoid Purified Agar) and overlay media (MEM, 0.3 % v/v BSA (fraction V), 2.8 nM L-Glutamine, 0.2 % v/v NaHCO$_2$, 14 mM HEPES, 0.001 % v/v Dextran, 0.1x Penicillin and 0.1x Streptomycin) containing 2 μgmL-1 TPCK trypsin. Agar was removed after 72 hr and cells fixed in 2 mL 4% paraformaldehyde in PBS for 1 hr prior to staining with 1 % v/v crystal violet solution for 5 min, enabling plaques to be visualised and counted.

## Lentiviral pseudotype entry assays

Murine leukaemia virus (MLV) based Lentiviral pseudotypes were generated using a three plasmid system comprising a luciferase reporter vector (pTG126), an+ MLV Gag-Pol expressing plasmid (phCMV-5349) and a plasmid encoding the vesicular stomatitis virus glycoprotein (VSV-G) as a positive control, relevant HCV E1/E2 sequences (pcDNA3.1D-E1/E2), or lacking an envelope as a negative control. Two mg of each plasmid were mixed and transfected using polyethylenimine (PEI) into HEK293T cells, seeded the previous day in a 10 cm culture dish ($1.2 \times 10^6$ per dish). Transfections were removed after 6 hr and media replaced (Dulbecco's Modified Eagle Medium with 10 % v/v foetal calf serum and non-essential amino acids). Supernatants were harvested at 72 hr post-transfection and clarified through a 0.45 μm filter prior to use.

Pseudotypes were generated using VSV-G, HCV envelopes from prototypic strains (genotype 1a (H77) and genotype 2a (JFH-1 and J6)), as well as three patient isolates UKNP1.18.1 (genotype 1b), UKNP3.2.1 and UKNP3.1.1 (both genotype 3a) as described previously (*Urbanowicz et al., 2016b*). Pseudotypes were treated with the indicated final concentrations of inhibitor (JK3/32, R21 or GNA) for 90 min prior to transduction of Huh7 cells in DMEM for 4 hr at 37°C. Following transduction, the media/pseudotype/inhibitor mix was removed and cells incubated in fresh DMEM for 72 hr before lysing and measuring luciferase activity in the HCVpp treated cells. Assays were normalised to positive controls lacking inhibitor and baseline determined using a pseudotype preparation with no E1/E2 present (delta-E). All conditions were performed in triplicate and data shown are representative of two independent experiments.

## Virus purification and ultracentrifugation

High titre J4/JFH-1 virus stocks were generated by sequential daily harvest of Huh7 culture supernatants over a 1 week period, comprising 20 mL of HEPES-buffered media in each of eight T150 cell culture flasks, seeded with $2 \times 10^6$ electroporated cells on day 1. Supernatants were clarified prior to addition of 1/3$^{rd}$ volume of 30% (w/v) polyethylene glycol (PEG)−8000 in PBS, thorough mixing and incubation at 4°C overnight. The next day, precipitates were spun at 2000 rpm for 40 min at 4°C in a Hereas bench-top laboratory centrifuge, pellets harvested, and then resuspended in 1/100$^{th}$ the original culture volume of PBS. Stocks were titred using the IncuCyte and snap-frozen in dry ice/ethanol prior to storage at −80°C.

Approximately $2 \times 10^6$ IU of virus were diluted into 200 µL PBS and treated with either DMSO or p7i at a final concentration of 10 µM. Suspensions were then layered over a pre-formed 10–40% (v/v) iodixinol gradient in a 2.2 mL open-topped mini-ultracentrifuge tube. Gradients were then centrifuged at 150 000 x $g$ for 3 hr at 4°C in a S55S Sorvall rotor prior to harvesting into twelve equal fractions. 10 µL was removed for infectivity testing and 50 µL for fluorimetry after adjusting to 0.1% (v/v) Triton-X100 to lyse virions. Blank gradients run in parallel were fractionated as above, then 100 µL per fraction was placed in labelled pre-weighed 1.5 mL Eppendorf tubes and resultant mass determined for density calculation. The remainder of gradient fractions were mixed with an equal volume of 2 x Laemmli SDS-PAGE sample buffer and stored at −20°C prior to thawing and analysis of 10 µL/fraction by SDS-PAGE and western blotting.

## SDS-PAGE and western blotting

Gradient samples were separated on a 4–20% Tris-Glycine acrylamide gel using a BioRad MiniProtean III rig, at a set voltage of 120 V for 60–90 min. Gels were then dismantled and placed within a pre-wetted sandwich of thick blotting paper on top of a pre-cut PVDF membrane (0.45 µm) that had been activated in 100% MeOH for 5 min at RT. Proteins were transferred for 2 hr using a Hoeffer semi-dry transfer rig set at 320 mA. Blotting sandwiches, gels and membranes were thoroughly pre-soaked in transfer buffer (1 x Tris-Glycine pH 8.3, 20% MeOH) prior to assembly. Membranes were removed from transfers, and placed in 20 mL blocking solution (5 % w/v fat-free milk in 1 x Tris-buffered Saline, 0.1% Tween 20 (TBS-T)) and shaken at RT for at least 3–4 hr. Membranes were then washed in TBS-T and placed in 10 mL of blocking solution containing primary antibody (mouse anti-HCV core, C7-50, Thermo Fisher catalogue # MA1-080) diluted 1/1000, at 4°C overnight with gentle shaking. The next day, primary antibody was removed by three washes in 1 x TBS-T at RT for 10 min, followed by incubation with secondary antibody diluted 1/5000 in blocking solution (goat anti-mouse HRP conjugate, SIGMA) for 2 hr at RT. Washing was then repeated prior to visualisation using ECL prime chemiluminescence substrate (GE Healthcare/Amersham), according to the manufacturer's instructions.

## EGF uptake assay

Huh7 cells were seeded into 24-well plates ($1.5 \times 10^5$ per well) and left to adhere for ~6 hr. Cells were then treated using titrated compounds or Bafilomycin A (1 µM) as a positive control for 4 hr. Cells were then washed extensively and media replaced with PBS containing 2 µgµL$^{-1}$ FITC-conjugated epidermal growth factor (EGF, Invitrogen) for 30 min. Cells were then washed extensively, removed using a cell scraper and then fixed in 4% (w/v) paraformaldehyde in PBS for 10 min at room temperature. Flow cytometry was then used to determine median FITC levels, with gating on intact cells. All conditions were performed in triplicate.

## Commercial kinase activity screen

JK3/32 was tested at the MRC Protein Phosphorylation Unit International Centre for Kinase Profiling, Dundee, Scotland (http://www.kinase-screen.mrc.ac.uk/services/express-screen). The express screen was undertaken (50 human kinases) using 10 µM JK3/32 in a highly accurate radioactive filter binding assay using $^{33}$P ATP. This method is sensitive, accurate and provides a direct measure of activity. A control plate using reference compounds is analysed in parallel for quality control purposes. Values for % kinase activity are then determined.

## Analysis of p7 sequence conservation

1456 aligned HCV p7 sequences from all genotypes were obtained from the EU HCV database website in FASTA format. Sequences were aligned and visualised using Jalview (www.jalview.org, RRID: SCR_006459), allowing the percentage occupancy at each of the 63 amino acid positions within the protein to be quantified (*Figure 2—figure supplement 1*). Data was exported into MS Excel, then inputted into a free sequence logo website (https://weblogo.berkeley.edu/) to visualise relative occupancy (*Figure 2d*).

## Prototypic p7i

Rimantadine hydrochloride was purchased from Chembridge, amantadine hydrochloride from SIGMA and NN-DNJ from Toronto Biochemicals. BIT225 was synthesised via 5-bromo-2-naphthoic acid and 5-(1-methyl-1H-pyrazol-4-yl)−2-naphthoic acid according to the patent detail (US20150023921A1); analytical data was consistent with the expected structure.

## Generation of optimised heptameric 3ZD0-based p7 channel structure and structure homology models

The initial heptameric channel model was constructed using the Maestro programme (Schrödinger, RRID:SCR_016748) based upon the monomeric unit from the 3ZD0 NMR structure as described (*Foster et al., 2014*). A heptameric bundle arrangement of protomers was oriented with His17 oriented towards the lumen equidistant from a centroid atom placed in the middle of the lumen to serve as a rotational centre with multiple energy minimisation steps. Iterative rotation of protomers in an octanol environment generated solutions. The preferred model was then refined using molecular dynamics simulations ( Materials and methods described below). Structure homology models for genotype 2a (JFH-1) and 3a (S52) were previously described (*Foster et al., 2014*).

## Virtual mutation and compound docking

For compound docking into Maestro generated models, the p7 structures were energy-minimised using the default Schrödinger's Protein Preparation Wizard (Version 12.3.13). Hydrogen atoms were added and partial charges were assigned. Minimization was carried out with an RMSD cutoff of 0.3 Å and using the OPLS3 force field to remove steric clashes and bad contacts. Ligands were prepared using Lig-Prep panel available in Schrödinger Maestro software with the OPLS3 force field at pH $7 \pm 2$, which generated low energetics 3D structures suitable for Glide-docking. Virtual mutation was carried out using the site-directed mutagenesis in Maestro. Each Tyr45 was virtually mutated to Thr, Leu50 to Phe and Leu51 to Cys. Finally, each mutated amino acid was minimised using the Minimise tool in Schrödinger Maestro with the OPLS3 force field and solvation model VSGB. For docking, the Leu20 residue was selected from each monomer of the wild type and mutated proteins respectively. The size of receptor grid box was $10 \times 8 \times 6$ Å. To investigate the interaction of the protein and ligands, the energy-minimised ligands were docked using Schrödinger's 'Ligand Docking' tool set as default settings. The binding energy ($\Delta G_{Bind}$) was estimated using the Prime MM-GBSA tool in Schrödinger Maestro with the OPLS3 force field and solvation model VSGB.

## Structure-guided molecular dynamics simulations and compound docking

### Molecular docking

The p7 structure was energy-minimised using the default energy minimization scheme in MOE software (Version 2015:1001, http://www.chemcomp.com, RRID:SCR_014882), AMBER 94 force field was used with $\varepsilon = 2$. Hydrogen atoms were added and partial charges were assigned using MOE. The energy minimization was carried out by restraining protein backbone atoms as rigid. The pairwise alignment of the structures, the one before minimization and the one after minimization, gives an RMSD of 0.10 Å.

Ligands were prepared using MOE software. Hydrogens atoms were added using the protonate 3D option and then partial charges were assigned with MMFF94 force field default parameters. Energy minimizations were performed using MMFF94 force field with a root mean square gradient of 0.1 kcal $Mol^{-1}.Å^{-1}$ and gas phase solvation model.

For docking, Leu20 or Phe20 residues were selected from each monomer of the wild type and mutated proteins respectively. Receptor sites with a radius of 5 Å is defined around Leu20 and Phe20 residue of each chain. The energy-minimised ligands were loaded into the MOE graphical user interface. The 'rigid receptor' protocol of MOE was used, where side chains of the protein were kept 'fixed' during the force field base refinement. Ligand placement was performed using the Triangle Matcher protocol, where the active site was defined using the α-spheres and the centre of that spheres are triplets of atoms. These triplets represent the location of tight packing. Poses are generated by superposing the ligand atom triplets onto the centre of the α-spheres. The top 1000 poses received after the placement were then used to score using the London dG scoring function which is an energy term summing contributions from ligand entropy from its topology or ligand flexibility, geometric imperfections of protein-ligand and metal-ligand interactions and desolvation energy. The top 30 poses ranked accordingly based on the London dG scores are then taken for a forcefield based refinement within the rigid receptor. The resulting poses are then rescored using Generalized Born Volume Integral/Weighted Surface Area dG (GBVI/WSA dG) scoring function. At the end final 30 poses were ranked accordingly whilst removing the duplicate poses.

## MD simulations

To simulate all the systems, Amber ff99SB-ILDN force field (ff) with Amber/Berger combination of ff was used with Gromacs 4.5.5. The POPC lipid (1-palmitoyl-2-oleoyl-sn-glycero-3-phosphocholine) topology was received from *Cordomí et al., 2012* whilst for the calculation of ligand parameters the generalized amber force field (GAFF) was used. The partial charges (ESP) were estimated using a HF/6–31G* basis set and Antechamber package was used for restrained electrostatic potential (RESP) fitting.

Each of the bundles, (wild type docked with JK3/32 and rimantadine, L20F mutated bundle docked with JK3/32 and rimantadine, and only protein) was inserted into patches of hydrated and pre-equilibrated POPC lipids. The overlapped lipids were removed and after steps of minimization (2000 steps of steepest decent and 5000 steps of conjugated gradient), it was equilibrated for a total of 1.9 ns by restraining the positions of the heavy atoms proteins and ligands. The systems were brought to equilibrium by gradually increasing the temperature of the systems from 100 K to 200 K and 310 K. During the initial equilibration, the peptides and ligands were fully restrained by applying a harmonic potential with a force constant, $k = 1000$ kJ mol$^{-1}$ nm$^{-2}$. The simulations at temperatures 100 K and 200 K were run for 200 ps each followed by simulation at 310 K for 500ps. Once the systems were equilibrated at 310K, the restraints were released by running three short (500 ps) simulations, one with $k = 500$ kJ mol$^{-1}$ nm$^{-2}$ the consecutive two simulations with $k = 250$ kJmol$^{-1}$ nm$^{-2}$ and finally 0 kJmol$^{-1}$ nm$^{-2}$.

For all the simulations SPC/E water model was employed and ion parameters proposed by Joung et al. were adopted (*Joung and Cheatham, 2009*). A Nosé-Hoover thermostat with a coupling time of 0.1 ps coupling separately to the temperature of the peptide, lipid, and the water molecules and Berendsen barostat with a coupling time of 2.0 ps were used during the MD simulations.

For the systems without any docked ligand and with JK3/32 docked, 14 Cl- ions were added to neutralize the simulation box whilst for the systems with rimantadine docked it was 15 Cl-ions. All the systems were consisted of 449 lipids which were hydrated with 14605–14606 water molecules (with 14 and 15 Cl-ions respectively).

## Compound synthesis and purification - general

Reagents and solvents were obtained from commercial suppliers and used without further purification. Thin layer chromatography (TLC) analyses were conducted using silica gel (aluminium foil backing) plates and visualised under UV radiation in a dark-box. Compound purification was effected using gradient elution on a Biotage Isolera-4 running SiO$_2$ cartridges. HPLC-MS was performed on a Bruker Daltronics spectrometer running a gradient of increasing acetonitrile (0 to 100%) in water containing 0.1% TFA at 1 mL.min$^{-1}$ on a short path $^{18}$C reverse phase column detecting compounds with both a diode array detector and a Bruker Mass spectrum analyser. HRMS experiments were conducted on a Bruker MaxisImpact time-of-flight spectrometer operating in a positive ion mode with sodium formate as an internal standard. $^1$H, $^{13}$C experiments were recorded using either a Bruker DRX 500 instrument or a Bruker DPX 300 operating at 298K, by solubilising the sample in

deuterated chloroform (CDCl$_3$) with internal standard tetramethylsilane (TMS), CD$_3$OD, d$_6$-DMSO, or d$_6$-acetone as the NMR solvent. Chemical shifts were expressed as parts per million (ppm) and the splitting signals of NMR assigned as s (singlet), d (doublet), t (triplet), dd (doublet of doublet), br (broad) or m (multiplet).

Compound syntheses and purification – specific schemes. (Please see Source Data file: 'Supp-CompoundStructures' for corresponding chemical structures).

## Synthesis of alexafluor-JK3/32 adduct (JK3/32-488)

### 3-[(dimethylamino)methylidene]−2,3-dihydro-1H-indol-2-one

Dimethylformamide dimethylacetal (1.2 g, 10 mmol) was added with cooling to a suspension of oxindole (1.30 g, 9.80 mmol) in chloroform (15 mL) before the contents were stirred at room temperature for 15 mins, and heated under reflux for 4 hr. The contents were cooled to room temperature and concentrated under reduced pressure to afford an orange solid, which was recrystallised from ethanol-diethyl ether to afford a yellow solid, which was used crude in the subsequent reaction.

### 3-[(dimethylamino)methylidene]−1-[(4-fluorophenyl)methyl]−2,3-dihydro-1H-indol-2-one

Caesium Carbonate (580 mg, 1.8 mmol) was added in a single portion with stirring to an acetonitrile (5 mL) suspension of 3-[(dimethylamino)methylidene]−2,3-dihydro-1H-indol-2-one (200 mg, 1.1 mmol) at room temperature and the contents were stirred for a further 30 min, prior to the addition of 4-fluorobenzylbromide (300 mg, 1.6 mmol) *via* a syringe, after which the contents were stirred for 4 hr at room temperature and then heated at 70°C for a further 1 hr. The reaction mixture was cooled to room temperature, and transferred to a separating funnel, whereupon the reaction mixture was diluted with water and extracted into dichloromethane (2 × 15 mL). After drying over sodium sulphate, the contents were filtered and the organic phase concentrated under reduced pressure to yellow oil, which was chromatographed (SiO$_2$; gradient elution; Hexane-EtOAc = 1: 1% to 100% EtOAc) to afford the desired product as pale yellow solid (210 mg, 67%) which was carried forward to the next step without additional characterisation.

### (3Z)−1-[(4-fluorophenyl)methyl]−3-({[4-(prop-2-yn-1-yloxy)phenyl]amino}methylidene)−2,3-dihydro-1H-indol-2-one (1191−146)

An ethanol (3 mL) suspension of 3-[(dimethylamino)methylidene]−1-[(4-fluorophenyl)methyl]−2,3-dihydro-1H-indol-2-one (53 mg, 0.18 mmol) was treated with 4-aminophenylpropargylether (35 mg, 0.23 mmol) and *p*-toluenesulphonic acid (52 mg, 0.28 mmol), and the contents were heated under reflux with stirring for 18 hr until HPLC-MS had indicated consumption of the dimethylenamide. Upon gradual cooling to 60°C, together with intermittent scratching of the inside of the flask with a micro-spatula, there was obtained a precipitate that was filtered at 50°C, washed with cold ethanol and recrystallised from chloroform-diethyl ether to yield the desired product as a yellow solid (26 mg, 37%). IR: v$_{max}$/cm$^{-1}$ (solid): 3080, 1633, 1610. HPLC-MS: 2.24 min, 399.5 [M+H]$^+$. $^1$H NMR (500 MHz, CDCl$_3$): 10.64 (d, *J* = 13 Hz, 1H), 7.87 (d, *J* = 13 Hz, 1H), 7.30 (d, *J* = 7 Hz, 1H), 7.21 (br, 3H), 7.06 (d, *J* = 8.5 Hz, 2H), 6.93 (m, 4H), 6.71 (d, *J* = 8.5 Hz, 1H), 4.96 (s, 2H), 4.63 (d, *J* = 2.5 Hz, 2H), 2.47 (d, *J* = 2.5 Hz, 1H); $^{13}$C (300 MHz, CDCl$_3$): 157.6, 149.5, 137.7, 132.8, 132.3, 129.2, 124.0, 121.3, 117.7, 116.4, 116.0, 115.7, 108.5, 99.6, 98.6, 78.4, 75.7, 56.3, 42.5.

## Alexafluor-JK3/32-triazole (JK3/32-488)

Alexafluor-488 (0.5 mg) was supplied in a light-proof centrifuge tube, to which was added a *tert*butanol (0.2 mL) suspension of the (3Z)−1-[(4-fluorophenyl)methyl]−3-({[4-(prop-2-yn-1-yloxy)phenyl]amino}methylidene)−2,3-dihydro-1H-indol-2-one (1191−146) (0.25 mg), followed by a solution (0.1 mL) aqueous ascorbic acid (0.35 mg). The contents were vortexed for 30 mins, before an aqueous solution (0.05 mL) of copper sulphate (0.20 mg) was added, and the contents shaken overnight at 40°C. Following this, the contents were concentrated on a Genevac, to a residue which was re-suspended in DMSO (0.20 mL) and centrifuged to disperse the solutes.

## Synthesis of JK3/32

### 1-benzyl-3-[(dimethylamino)methylidene]−2,3-dihydro-1H-indol-2-one

A DMF (4 mL) suspension of (3$Z$)−3-[(dimethylamino)methylidene]−1,3-dihydro-2H-indol-2-one (300 mg, 1.6 mmol) was cooled to 0°C via an ice-bath with rapid stirring. Sodium hydride (125 mg, 60% dispersion) was added in three portions over 10 min, and the resulting yellow suspension stirred for a further 20 min at 5°C, prior to the addition of benzyl bromide (330 mg, 1.90 mmol), and the contents left to stir for a further 45 min and allowed to warm to 25°C over this period. Saturated ammonium chloride solution was added dropwise with cooling, and the contents transferred with ethyl acetate to a separating funnel whereupon the organic phase was removed, washed with water and brine, and dried over sodium sulphate. Evaporation and chromatography (SiO$_2$; gradient elution; hexane: EtOAc = 2: 1% to 100% EtOAc) afforded the desired product (210 mg, 47%) as an oil which solidified upon standing. HPLC-MS: 1.89 min, 279.1 [M+H]$^+$. IR: $v_{max}$/cm$^{-1}$ (solid): 1625: $^1$H NMR (500 MHz, CDCl$_3$): 3.34–3.37, (6H, m, 2CH$_3$), 5.06 (2H, s, CH$_2$Ph), 6.74–7.42 (8H, m, ArH), 7.49 (1H, d, H-4), 7.68 (1H, s, CH).

### 1-benzyl-3-[(4-methoxyphenyl)aminomethylidene]−2,3-dihydro-1H-indol-2-one (JK3/32)

An ethanol (3 mL) solution of p-toluenesulfonic acid (26 mg, 0.14 mmol) was combined with 4-anisidine (17 mg, 0.14 mmol) at room temperature, and sonicated briefly to disperse the contents. 1-benzyl-3-[(dimethylamino)methylidene]−2,3-dihydro-1H-indol-2-one (35 mg, 0.13 mmol) was added in a single portion and the contents heated under reflux with stirring for 12 hr to provide a precipitate, which was diluted with methanol (0.5 mL) and filtered at 50°C to afford the desired product as a yellow solid (18 mg, 39%). IR: $v_{max}$/cm$^{-1}$ (solid): 3070, 1631, 1610. HPLC-MS: 2.21 min, 357.1[M+H]$^+$. $^1$H NMR (500 MHz, CDCl$_3$): δ 10.76 (d, $J$ = 13 Hz, 1H), 7.97 (d, $J$ = 13 Hz, 1H), 7.38 (d, $J$ = 6.5 Hz, 1H), 7.31 (m, 3H), 7.24 (m, 2H), 7.12 (d, $J$ = 8.5 Hz, 2H), 7.03 (m, 2H), 6.94 (d, $J$ = 8.5 Hz, 2H), 6.82 (d, $J$ = 8.0 Hz, 1H), 5.07 (s, 2H), 3.82 (s, 3H); $^{13}$C NMR (300 MHz, d$_6$-DMSO) δ 167.8, 156.1, 139.4, 137.5, 137.1, 133.4, 128.6, 127.3, 127.0, 123.4, 120.7, 118.9, 117.8, 116.7, 115.1, 108.4, 97.4, 55.6, 42.3. HRMS (m/z): [M+H]$^+$ calcd. for C$_{23}$H$_{20}$N$_2$O$_2$, 357.1598, found 357.1602.

### (3Z)−3-[(4-methoxyanilino)methylidene]−1-phenyl-1,3-dihydro-2H-indol-2-one (JK3-42)

To N-phenyloxindole (0.9 mmol) was added dimethylformamide dimethyl acetal (5 mL) and the mixture heated at 70°C for 1 hr. The mixture was concentrated, dissolved in ethanol (3 mL) and then p-toluenesulfonic acid (48 mg, 0.24 mmol) and 4-anisidine (30 mg, 0.24 mmol) added at room temperature and the contents heated under reflux with stirring for 12 hr to provide a precipitate which was diluted with methanol (0.5 mL) and filtered at 50°C to afford the desired product as a pale yellow solid (19 mg, 42%). IR: $v_{max}$/cm$^{-1}$ (solid): 3080, 1633, 1610. HPLC-MS: 2.13 min, 345.1[M+H]$^+$. $^1$H NMR (500 MHz, CDCl$_3$): δ 10.78 (d, $J$ = 13 Hz, 1H), 7.94 (d, $J$ = 13 Hz, 1H), 7.33 (d, $J$ = 6.9 Hz, 1H), 7.25 (m, 3H), 7.18 (m, 2H), 7.12 (d, $J$ = 8.3 Hz, 2H), 7.03 (m, 2H), 6.94 (d, $J$ = 8.3 Hz, 2H), 6.78 (d, $J$ = 8.3 Hz, 1H), 3.87 (s, 3H); $^{13}$C NMR (300 MHz, d$_6$-DMSO) δ 165.8, 158.1, 141.4, 138.5, 137.3, 135.1, 128.6, 127.6, 127.0, 125.9, 124.8, 123.1, 117.9, 116.7, 116.1, 109.3, 96.4, 55.7. HRMS (m/z): [M+H]$^+$ calcd. for C$_{22}$H$_{18}$N$_2$O$_2$, 345.1575, found 345.1601.

### (3Z)−3-[(4-methoxyanilino)methylidene]−1,3-dihydro-2H-indol-2-one (JK3-38)

An ethanol (3 mL) solution of p-toluenesulfonic acid (26 mg, 0.14 mmol) was combined with 4-anisidine (17 mg, 0.14 mmol) at room temperature, and sonicated briefly to disperse the contents. (3$Z$)−3-[(dimethylamino)methylidene]−1,3-dihydro-2$H$-indol-2-one (0.14 mmol) was added in a single portion and the contents heated under reflux with stirring for 12 hr to provide a precipitate, which was diluted with methanol (0.5 mL) and filtered at 50°C to afford the desired product as a yellow solid (15 mg, 39%). IR: $v_{max}$/cm$^{-1}$ (solid): 3100, 3050, 1640, 1610. HPLC-MS: 1.97 min, 266.1[M+H]$^+$. $^1$H NMR (500 MHz, CDCl$_3$): δ 10.50 (d, $J$ = 11.0 Hz, 1H), 9.99 (s, 1H), 8.08 (d, $J$ = 11.0 Hz, 1H), 7.55 (d, $J$ = 6.9 Hz, 1H), 7.25 (m, 2H), 6.85–6.92 (4H, m), 6.78 (d, $J$ = 8.3 Hz, 1H), 3.87 (s, 3H); $^{13}$C NMR (300

MHz, $d_6$-DMSO) δ 165.8, 158.1, 141.4, 138.5, 135.1, 129.6, 126.0, 124.9, 124.0, 123.1, 117.5, 116.0, 115.1, 107.3, 93.4, 55.7. HRMS (m/z): [M+H]$^+$ calcd. for $C_{16}H_{14}N_2O_2$, 266.1102, found 266.1197.

### (Z)−1-benzyl-3-(2-(4-methoxyphenyl)hydrazono)indolin-2-one (21-RS-7)

To a solution of *N*-benzyl isatin (0.05 g, 0.21 mmol) in ethanol (5 mL) was added *p*-methoxyphenylhydrazine hydrochloride (0.3 mL, 0.21 mmol) and the mixture heated at reflux for 2 hr. The reaction mixture was cooled to ambient temperature and the resultant precipitate filtered and dried to give the title compound as a yellow powder (0.04 g, 0.11 mmol, 50%). IR: $v_{max}$/cm$^{-1}$ (solid): 1663, 1610. HPLC-MS (ES): 2.42 min, m/z = 737.6 (2M+Na)$^+$ $^1$H NMR (300 MHz, DMSO-$d_6$): δ 12.75 (s, 1H), 7.59 (dd, *J* = 7.2, 0.9 Hz 1H), 7.46 (d, *J* = 9.0 Hz, 2H), 7.38–7.25 (m, 5H), 7.22 (dd, *J* = 7.7, 1.3 Hz, 1H), 7.15–7.01 (m, 2H), 6.99 (d, *J* = 9.0 Hz, 2H), 5.03 (s, 2H), 3.76 (s, 3H) ppm; $^{13}$C NMR (300 MHz, CDCl$_3$): δ 162.3, 156.2, 139.8, 136.4, 135.9, 128.8, 127.7, 127.4, 127.3, 125.6, 122.5, 121.6, 118.6, 115.7, 114.8, 109.2, 55.6, 43.2 ppm;; HRMS (m/z): [M+H]$^+$ calcd for $C_{22}H_{19}N_3O_2Na$ requires 380.1369 Found: 380.1374 (M+Na)$^+$: $v_{max}$/cm$^{-1}$ (solid): 1663, 1610; M.pt: 145–147°C.

### (Z)−1-benzyl-3-(2-phenylhydrazono)indolin-2-one (21-RS-8)

To a solution of *N*-benzyl isatin (0.05 g, 0.21 mmol) in ethanol (5 mL) was added phenylhydrazine hydrochloride (0.03 mg, 0.21 mmol) and the mixture heated at reflux for 2 hr. The reaction mixture was cooled to ambient temperature and the resultant precipitate filtered and dried to give the title compound as a yellow powder (0.03 g, 0.09 mmol, 42%).

IR: $v_{max}$/cm$^{-1}$ (solid): 3060, 1663. $^1$H NMR (300 MHz, DMSO-$d_6$): δ 12.72 (s, 1H), 7.64–7.59 (m, 1H), 7.52–7.46 (m, 2H), 7.43–7.30 (m, 6H), 7.30–7.22 (m, 2H), 7.16–7.07 (m, 1H), 7.10–7.01 (m, 2H), 5.03 (s, 2H) ppm; $^{13}$C NMR (300 MHz, DMSO-$d_6$): 161.1, 142.4, 140.0, 136.3, 129.5, 128.7, 128.3, 127.5, 127.4, 126.5, 123.1, 122.5, 120.5, 118.5, 114.3, 109.8 ppm; HPLC-MS (ES): 2.07 min, m/z = 677.7 (2M+Na)$^+$; HRMS (m/z): [M+H]$^+$ calcd for $C_{21}H_{17}N_3ONa$ requires 350.1264, Found: 350.1267 (M+Na)$^+$,.; HPLC: RT = 4.51 min (100%); M.pt: 134–136°C.

### (Z)−3-(2-(4-methoxyphenyl)hydrazono)−1-phenylindolin-2-one (21-RS-9)

To a solution of *N*-phenyl isatin (0.05 g, 0.22 mmol) in ethanol (5 mL) was added *p*-methoxyphenylhydrazine hydrochloride (0.04 mg, 0.22 mmol) and the mixture heated at reflux for 2 hr. The reaction mixture was cooled to ambient temperature and the resultant precipitate filtered and dried to give the title compound as a yellow powder (0.06 g, 0.17 mmol, 77%).

IR: $v_{max}$/cm$^{-1}$ (solid): 3186, 1673. $^1$H NMR (300 MHz, CDCl$_3$): δ 12.88 (s, 1H), 7.69–7.60 (m, 1H), 7.52–7.45 (m, 2H), 7.43–7.33 (m, 4H), 7.26 (d, *J* = 9.0 Hz, 2H), 7.17–7.05 (m, 2H), 6.86 (d, *J* = 8.9 Hz, 2H), 3.75 (s, 3H) ppm; $^{13}$C NMR (300 MHz, CDCl$_3$): δ 161.6, 156.2, 140.3, 136.3, 133.9, 129.6, 128.1, 127.4, 126.4, 125.3, 123.0, 121.6, 118.6, 115.7, 114.8, 109.7, 55.6 ppm; HPLC-MS (ES): : RT = 2.46 min, m/z = 709.9 (2M+Na)$^+$; HRMS (m/z): [M+H]$^+$ calcd for $C_{21}H_{17}N_3O_2Na$ requires 366.1209: 366.1209 (M+Na)$^+$,;;; M.pt: 195–197°C.

### 1-benzyl-3-(2-(3,4-dimethoxyphenyl)−2-oxoethyl)−3-hydroxyindolin-2-one (21-RS-11 (aka 'R21'))

To a microwave vial was charged *N*-benzyl isatin (0.20 g, 0.84 mmol), 3,4-dimethoxyacetophenone (0.17 g, 0.93 mmol), diethylamine (two drops) and ethanol (2 mL). The reaction was heated at 100°C and 300 W within a microwave reactor for 1–2 hr. The crude product was purified *via* column chromatography (50:50 hexane:ethyl acetate) to yield the <u>title compound</u> as a cream powder (0.24 g, 0.58 mmol, 68%). IR: $v_{max}$/cm$^{-1}$ (solid): 3329, 1668, 1598. $^1$H NMR (300 MHz, CDCl$_3$): δ 7.54 (d, *J* = 8.4 Hz, 1H), 7.48 (s, 1H), 7.43 (d, *J* = 7.4 Hz), 7.40–7.29 (m, 5H), 7.21 (t, *J* = 7.8 Hz, 1H), 7.01 (t, *J* = 7.5 Hz, 1H), 6.86 (d, *J* = 8.4 Hz, 1H), 6.74 (d, *J* = 7.8 Hz, 1H), 4.95 (s, 2H), 4.83 (br s, 1H), 3.95 (s, 3H), 3.91 (s, 2H), 3.86 (d, *J* = 17.6 Hz, 1H), 3.59 (d, *J* = 17.1 Hz, 1H) ppm; $^{13}$C NMR (300 MHz, CDCl$_3$): δ 196.8, 176.6, 153.9, 149.0, 142.8, 135.5, 130.2, 129.8, 129.6, 128.8, 127.7, 127.3, 124.1, 123.3, 123.1, 110.0, 109.9, 109.7, 80.5, 74.7, 56.2, 56.0, 43.9 ppm; HPLC-MS (ES): RT = 1.87 min, m/z = 857.6 (2M+Na)$^+$; HRMS (m/z): [M+H]$^+$ calcd for $C_{25}H_{23}NO_5Na$ requires 440.1468, Found: 440.1488 (M+Na)$^+$. M.pt: 157–159°C.

### 1-benzyl-3-(2-(3,4-dimethoxyphenyl)−2-oxoethylidene)indolin-2-one (21-RS-17)

To a stirred solution of 1-benzyl-3-(2-(3,4-dimethoxyphenyl)−2-oxoethyl)-3-hydroxyindolin-2-one (21-RS-11) (0.04 g, 0.08 mmol) in acetic acid (5 mL) was added 12M aqueous hydrochloric acid (2 mL). The mixture was heated at 80°C for 0.5 hr then stirred at ambient temperature for 2 days. The mixture was quenched with cold aqueous sodium bicarbonate (20 mL), extracted with ethyl acetate (3 × 20 mL) and concentrated to dryness. The crude product was purified via column chromatography (50:50 hexane:ethyl acetate) to give the title compound as an orange powder (0.02 g, 0.05 mmol, 57%). IR: $v_{max}/cm^{-1}$ (solid): 1705, 1599. $^1$H NMR (300 MHz, CDCl$_3$): δ 8.19 (d, $J$ = 7.6 Hz, 1H), 7.88 (s, 1H), 7.72 (dd, $J$ = 8.4, 2.0 Hz, 1H), 7.61 (d, $J$ = 2.0 Hz, 1H), 7.31–7.12 (m, 6H), 6.96–6.83 (m, 2H), 6.64 (d, $J$ = 7.8 Hz, 1H), 4.91 (s, 2H), 3.92 (d, $J$ = 2.4 Hz, 6H) ppm; $^{13}$C NMR (300 MHz, CDCl$_3$): δ 189.6, 168.2, 154.2, 149.5, 144.9, 135.6, 135.5, 132.2, 130.8, 128.9, 127.8, 127.6, 127.3, 127.2, 124.4, 122.8, 110.1, 109.2, 105.0, 80.2, 56.2, 56.1, 43.9 ppm; HPLC-MS (ES): RT = 2.18 min, m/z = 00.1 (M+H)$^+$; HRMS (m/z): [M+H]$^+$ calcd for C$_{25}$H$_{21}$NO$_4$Na requires 422.1363 Found: 422.1356 (M+Na)$^+$,; M.pt: 146–148°C.

### (3Z)−1-benzyl-3-[(2-methoxyanilino)methylidene]−1,3-dihydro-2H-indol-2-one (1191−104)

An ethanol (3 mL) solution of p-toluenesulfonic acid (52 mg, 0.27 mmol) was combined with 4-anisidine (34 mg, 0.27 mmol) at room temperature, and sonicated briefly to disperse the contents. 1-benzyl-3-[(dimethylamino)methylidene]−2,3-dihydro-1H-indol-2-one (68 mg, 0.27 mmol) was added in a single portion and the contents heated under reflux with stirring for 12 hr to provide a precipitate, which was diluted with methanol (0.5 mL) and filtered at 50°C to afford the desired product as a yellow solid (70 mg, 75%). IR: $v_{max}/cm^{-1}$ (solid): 3079, 1630. HPLC-MS: 2.28 min, 357.1 [M+H]$^+$. $^1$H NMR (500 MHz, CDCl$_3$): δ 11.02 (d, $J$ = 12.5 Hz, 1H), 8.10 (d, $J$ = 12.5 Hz, 1H), 7.43 (d, $J$ = 8.5 Hz, 1H), 7.32 (m, 6H), 7.05 (m, 4H), 6.99 (d, $J$ = 8.0 Hz, 1H), 6.80 (d, $J$ = 7.8 Hz, 1 hr), 5.12 (s, 2H), 4.09 (s, 3H); $^{13}$C NMR (125 MHz, CDCl$_3$) 157.6, 149.1, 132.7, 131.7, 130.4, 125.8, 123.5, 122.0, 119.2, 118.3, 115.7, 111.8, 110.8, 107.5, 105.9, 103.8, 97.2, 50.7, 37.8.

### 4-{[(Z)-(1-benzyl-2-oxo-1,2-dihydro-3H-indol-3-ylidene)methyl]amino}benzonitrile (1191−112)

An ethanol (3 mL) solution of p-toluenesulfonic acid (52 mg, 0.27 mmol) was combined with 4-aminophenylnitrile (33 mg, 0.27 mmol) at room temperature. 1-benzyl-3-[(dimethylamino)methylidene]−2,3-dihydro-1H-indol-2-one (68 mg, 0.27 mmol) was added in a single portion and the contents heated under reflux with stirring for 12 hr to provide a precipitate, which was diluted with methanol (0.5 mL) and filtered at 50°C to afford the desired product as a yellow solid (50 mg, 52%). IR: $v_{max}/cm^{-1}$ (solid): 3079, 2127, 1630. HPLC-MS: 2.20 min, 352.5 [M+H]$^+$. $^1$H NMR (500 MHz, CDCl$_3$): δ 10.89 (d, $J$ = 13.5 Hz, 1H), 7.89 (d, J = 12.8 Hz, 1H), 7.58 (d, $J$ = 7.5 Hz, 2H), 7.35 (d, $J$ = 9.0 Hz, 1H), 7.24 (m, 4H), 7.18 (m, 1H), 7.12 (d, $J$ = 7.5 Hz, 2H), 7.04 (t, $J$ = 7.5 Hz, 1H), 6.97 (t, $J$ = 6.0 Hz, 1H), 6.75 (d, $J$ = 6.0 Hz, 1H), 4.92 (s, 2H); $^{13}$C NMR (125 MHz, CDCl$_3$) 153.8, 143.6, 134.1, 130.8, 128.9, 127.7, 127.2, 125.7, 122.6, 121.8, 118.6, 116.9, 115.1, 109.16, 105.7, 104.6, 93.1, 43.4.

### (3Z)−1-benzyl-6-fluoro-3-[(4-methoxyanilino)methylidene]−1,3-dihydro-2H-indol-2-one (1191−121)

A DMF (4 mL) suspension of (3Z)−3-[(dimethylamino)methylidene]−6-fluoro-1,3-dihydro-2H-indol-2-one (300 mg, 1.60 mmol) was cooled to 0°C via an ice-bath with rapid stirring. Sodium hydride (125 mg, 60% dispersion) was added in three portions over 10 min, and the resulting yellow suspension stirred for a further 20 min at 5°C, prior to the addition of benzyl bromide (1.80 mmol) and the contents left to stir for a further 45 min and allowed to warm to 25°C over this period. Saturated ammonium chloride solution was added dropwise with cooling, and the contents transferred with ethyl acetate to a separating funnel whereupon the organic phase was removed, washed with water and brine, and dried over sodium sulphate. Evaporation and chromatography (SiO$_2$; gradient elution; hexane: EtOAc = 2: 1% to 100% EtOAc) afforded the desired product as an oil which solidified upon standing. This was then added to an ethanol (3 mL) solution of p-toluenesulfonic acid (52 mg, 0.27

mmol) and 4-anisidine (34 mg, 0.27 mmol) and the contents heated under reflux with stirring for 12 hr to provide a precipitate, which was diluted with methanol (0.5 mL) and filtered at 50°C to afford the desired product as a pale yellow solid (45 mg, 48%). IR: $v_{max}/cm^{-1}$ (solid): 3099, 1630. HPLC-MS: 2.24 min, 375.3 $[M+H]^+$. $^1H$ NMR (500 MHz, CDCl$_3$): δ 10.79 (d, $J$ = 13.5 Hz, 1H), 7.92 (d, $J$ = 13.5 Hz, 1H), 7.28 (m, 5H), 7.12 (d, $J$ = 10 Hz, 2H), 7.07 (m, 1H), 6.93 (d, $J$ = 8.5 Hz, 2H), 6.70 (m, 2H), 5.05 (s, 2H), 3.82 (s, 3H); $^{13}C$ NMR (300 MHz, CDCl$_3$): 155.4, 148.1, 138.6, 133.5, 132.2, 128.7, 127.7, 127.4, 127.2, 126.0, 125.2, 118.1, 115.3, 109.9, 108.9, 103.3, 99.2, 55.8, 43.8.

### 4-({[(3Z)−5-fluoro-3-[(4-methoxyanilino)methylidene]−2-oxo-2,3-dihydro-1H-indol-1-yl}methyl)benzonitrile (1191–120)

A DMF (4 mL) suspension of (3$Z$)−3-[(dimethylamino)methylidene]−5-fluoro-1,3-dihydro-2$H$-indol-2-one (300 mg, 1.6 mmol) was cooled to 0°C via an ice-bath with rapid stirring. Sodium hydride (125 mg, 60% dispersion) was added in three portions over 10 min, and the resulting yellow suspension stirred for a further 20 min at 5°C, prior to the addition of 4-cyanophenylmethyl bromide (369 mg, 1.90 mmol) and the contents left to stir for a further 45 min and allowed to warm to 25°C over this period. Saturated ammonium chloride solution was added dropwise with cooling, and the contents transferred with ethyl acetate to a separating funnel whereupon the organic phase was removed, washed with water and brine, and dried over sodium sulphate. Evaporation and chromatography (SiO$_2$; gradient elution; hexane: EtOAc = 2: 1% to 100% EtOAc) afforded the desired product as an oil. This was then added to an ethanol (3 mL) solution of $p$-toluenesulfonic acid (52 mg, 0.27 mmol) and 4-anisidine (34 mg, 0.27 mmol) at room temperature and the contents heated under reflux with stirring for 12 hr to provide a precipitate which was diluted with methanol (0.5 mL) and filtered at 50°C to afford the desired product as a yellow solid (51 mg, 48%). IR: $v_{max}/cm^{-1}$ (solid): 3130, 2145, 1620. HPLC-MS: 2.02 min, 400.4 $[M+H]^+$. $^1H$ NMR (500 MHz, CDCl$_3$): δ 10.75 (d, $J$ = 13 Hz, 1H), 7.95 (d, $J$ = 13 Hz, 1H), 7.61 (d, $J$ = 10 Hz, 2H), 7.38 (d, $J$ = 8.5 Hz, 2H), 7.12 (d, $J$ = 10 Hz, 2H), 7.09 (m, 1H), 6.94 (d, $J$ = 8.5 Hz, 2H), 6.73 (t, $J$ = 7.5 Hz, 1H), 6.60 (dd, $J$ = 8.5 Hz, 4.2 Hz, 1H), 5.1 (s, 2H), 3.82 (s, 3H); $^{13}C$ NMR (300 MHz, CDCl$_3$) 169.1, 160.3, 157.0, 142.2, 139.1, 133.1, 132.9, 127.9, 118.7, 118.1, 115.2, 111.6, 110.4, 108.6, 55.9, 43.1.

### (3Z)−5-fluoro-1-[(4-fluorophenyl)methyl]−3-[(4-methoxyanilino)methylidene]−1,3-dihydro-2H-indol-2-one (1191–124)

A DMF (4 mL) suspension of (3$Z$)−3-[(dimethylamino)methylidene]−5-fluoro-1,3-dihydro-2$H$-indol-2-one (300 mg, 1.6 mmol) was cooled to 0°C. Sodium hydride (125 mg, 60% dispersion) was added in three portions over 10 min, and the resulting yellow suspension stirred for a further 20 min at 5°C, prior to the addition of 4-fluorophenylmethyl bromide (1.80 mmol), and the contents left to stir for a further 45 min. Saturated ammonium chloride solution was added dropwise with cooling, and the contents transferred with ethyl acetate to a separating funnel whereupon the organic phase was removed, washed with water and brine, and dried over sodium sulphate. Evaporation and chromatography (SiO$_2$; gradient elution; hexane: EtOAc = 2: 1% to 100% EtOAc) afforded the desired intermediate as an oil. This was then added to an ethanol (3 mL) solution of $p$-toluenesulfonic acid (52 mg, 0.27 mmol) and 4-anisidine (34 mg, 0.27 mmol) at room temperature and the contents heated under reflux with stirring for 12 hr to provide a precipitate which was diluted with methanol (0.5 mL) and filtered at 50°C to afford the desired product as a yellow solid (51 mg, 53%). IR: $v_{max}/cm^{-1}$ (solid): 3115, 1625. HPLC-MS: 2.26 min, 393.2 $[M+H]^+$. $^1H$ NMR (500 MHz, CDCl$_3$): δ 10.71 (d, $J$ = 13.5 Hz, 1H), 7.86 (d, $J$ = 13.0 Hz, 1H), 7.19 (br, 1H), 7.05 (d, $J$ = 8.5 Hz, 2H), 6.99 (d, $J$ = 8.5 Hz, 1H), 6.92 (m, 3H), 6.87 (d, $J$ = 7.7 Hz, 2H), 6.66 (t, $J$ = 7.7 Hz, 1H), 6.59 (br, 1H), 4.93 (s, 2H), 3.76 (s, 3H); $^{13}C$ NMR (300 MHz, CDCl$_3$): 157.9, 149.6, 140.2, 138.0, 133.4, 133.2, 132.9, 128.9, 122.2, 118.0, 115.8, 115.6, 115.2, 110.2, 108.8, 103.4, 98.1, 55.6, 42.6.

### (3Z)−1-[(3,5-dimethyl-1,2-oxazol-4-yl)methyl]−3-[(4-methoxyanilino)methylidene]−1,3-dihydro-2H-indol-2-one (1191–106)

A DMF (4 mL) suspension of (3$Z$)−3-[(dimethylamino)methylidene]−1,3-dihydro-2$H$-indol-2-one (300 mg, 1.60 mmol) was cooled to 0°C via an ice-bath with rapid stirring. Sodium hydride (125 mg, 60% dispersion) was added in three portions over 10 min, and the resulting yellow suspension stirred for a further 20 min at 5°C, prior to the addition of 2,5-dimethylisoxazole-4-methylchloride (1.80 mmol)

and the contents left to stir for a further 45 min. Saturated ammonium chloride solution was added dropwise with cooling, and the contents transferred with ethyl acetate to a separating funnel whereupon the organic phase was removed, washed with water and brine, and dried over sodium sulphate. Evaporation and chromatography (SiO$_2$; gradient elution; hexane: EtOAc = 2: 1% to 100% EtOAc) afforded an intermediate as an oil. This was then added to an ethanol (3 mL) solution of *p*-toluenesulfonic acid (48 mg, 0.24 mmol) and 4-anisidine (30 mg, 0.24 mmol) at room temperature and the contents heated under reflux with stirring for 12 hr to provide a precipitate which was diluted with methanol (0.5 mL) and filtered at 50°C to afford the desired product as a pale yellow solid (57 mg, 54%). IR: ν$_{max}$/cm$^{-1}$ (solid): 3120, 1625. HPLC-MS: 1.91 min, 375.3 [M+H]$^+$. $^1$H NMR (500 MHz, CDCl$_3$): δ 10.67 (d, *J* = 13 Hz, 1H), 7.97 (d, *J* = 13 Hz, 1H), 7.72 (d, *J* = 10 Hz, 1H), 7.39 (d, *J* = 10 Hz, 1H), 7.15 (d, *J* = 10 Hz, 1H), 7.08 (m, 2H), 6.96 (d, *J* = 10 Hz, 2H), 6.74 (d, *J* = 8 Hz, 1H), 4.84 (s, 2H), 3.86 (s, 3H), 2.42 (s, 3H), 2.25 (s, 3H); $^{13}$C NMR (300 MHz, CDCl$_3$) 168.9, 166.9, 159.5, 156.7, 138.4, 137.0, 133.6, 129.0, 126.3, 123.9, 121.7, 117.9, 116.1, 115.3, 109.7, 108.5, 55.8, 32.5, 11.4, 10.6.

## (3Z)−3-[(4-methoxyanilino)methylidene]−1-(2-phenylethyl)−1,3-dihydro-2H-indol-2-one (1191−137)

A DMF (4 mL) suspension of (3*Z*)−3-[(dimethylamino)methylidene]−1,3-dihydro-2*H*-indol-2-one (300 mg, 1.6 mmol) was cooled to 0°C *via* an ice-bath with rapid stirring. Sodium hydride (125 mg, 60% dispersion) was added in three portions over 10 min, and the resulting yellow suspension stirred for a further 20 min at 5°C, prior to the addition of phenylethyl bromide (369 mg, 1.90 mmol), and the contents left to stir for a further 45 min and allowed to warm to 25°C over this period. Saturated ammonium chloride solution was added dropwise with cooling, and the contents transferred with ethyl acetate to a separating funnel whereupon the organic phase was removed, washed with water and brine, and dried over sodium sulphate. Evaporation and chromatography (SiO$_2$; gradient elution; hexane: EtOAc = 2: 1% to 100% EtOAc) afforded an intermediate product as an oil which solidified upon standing. The material was used immediately in the subsequent reaction. The oil was added to an ethanol (3 mL) solution of *p*-toluenesulfonic acid (53 mg, 0.27 mmol) and 4-anisidine (35 mg, 0.27 mmol) at room temperature and the contents heated under reflux with stirring for 12 hr to provide a precipitate which was diluted with methanol (0.5 mL) and filtered at 50°C to afford the desired product as a pale yellow solid (49 mg, 44%). IR: ν$_{max}$/cm$^{-1}$ (solid): 3095, 1620. HPLC-MS: 2.29 min, 371.2 [M+H]$^+$. $^1$H NMR (500 MHz, CDCl$_3$): δ 10.61 (d, *J* = 13 Hz, 1H), 7.86 (d, *J* = 13 Hz, 1H), 7.30 (d, *J* = 8.0 Hz, 1H), 7.22 (m, 3H), 7.17 (m, 2H), 7.03 (m, 3H), 6.99 (d, *J* = 7.0 Hz, 1H), 6.85 (d, *J* = 7.0 Hz, 2H), 6.81 (d, *J* = 8.0 Hz, 1H), 3.99 (t, *J* = 7.5 Hz, 2H), 3.77 (s, 3H), 2.95 (t, *J* = 7.5 Hz, 2H); $^{13}$C NMR (300 MHz, CDCl$_3$) 166.2, 149.5, 137.6, 133.7, 133.1, 131.4, 129.6, 128.9, 128.6, 128.4, 127.7, 121.1, 117.6, 117.5, 115.1, 108.4, 98.5, 55.8, 34.7, 33.5.

## (3Z)−1-[(4-fluorophenyl)methyl]−3-[(4-methoxyanilino)methylidene]−1,3-dihydro-2H-indol-2-one (1191−140)

A DMF (4 mL) suspension of (3*Z*)−3-[(dimethylamino)methylidene]−1,3-dihydro-2*H*-indol-2-one (300 mg, 1.6 mmol) was cooled to 0°C *via* an ice-bath with rapid stirring. Sodium hydride (125 mg, 60% dispersion) was added in three portions over 10 min, and the resulting yellow suspension stirred for a further 20 min at 5°C, prior to the addition of 4-flurophenylbromide (1.65 mmol), and the contents left to stir for a further 45 min. Saturated ammonium chloride solution was added dropwise and the contents transferred with ethyl acetate to a separating funnel whereupon the organic phase was removed, washed with water and brine, and dried over sodium sulphate. Evaporation and chromatography (SiO$_2$; gradient elution; hexane: EtOAc = 2: 1% to 100% EtOAc) afforded an oil which was added to ethanol (3 mL) and *p*-toluenesulfonic acid (53 mg, 0.27 mmol) and 4-anisidine (35 mg, 0.27 mmol) at room temperature and the contents heated under reflux with stirring for 12 hr to provide a precipitate which was diluted with methanol (0.5 mL) and filtered at 50°C to afford the desired product as a pale yellow solid (67 mg, 63%). IR: ν$_{max}$/cm$^{-1}$ (solid): 3105, 1635. HPLC-MS: 2.24 min, 375.3 [M+H]$^+$. $^1$H NMR (500 MHz, CDCl$_3$): δ 10.63 (d, *J* = 13 Hz, 1H), 7.87 (d, *J* = 13 Hz, 1H), 7.31 (d, *J* = 7.5 Hz, 1H), 7.21 (m, 2H), 7.04 (d, *J* = 9 Hz, 2H), 6.94 (m, 3H), 6.85 (d, *J* = 8.5 Hz, 2H), 6.72 (d, *J* = 8.0 Hz, 1H), 4.97 (s, 2H), 3.75 (s, 3H); $^{13}$C NMR (300 MHz, CDCl$_3$) 165.1, 156.5, 137.9, 137.4,

136.8, 135.3, 133.5, 132.6, 130.3, 128.8, 123.9, 123.8, 121.4, 117.1, 115.6, 115.1, 108.7, 98.1, 55.7, 42.7.

## (3Z)−3-[(4-fluoroanilino)methylidene]−1-[(4-fluorophenyl)methyl]−1,3-dihydro-2H-indol-2-one (1191−141)

A DMF (4 mL) suspension of (3Z)−3-[(dimethylamino)methylidene]−1,3-dihydro-2H-indol-2-one (300 mg, 1.6 mmol) was cooled to 0°C via an ice-bath with rapid stirring. Sodium hydride (125 mg, 60% dispersion) was added in three portions over 10 min, and the resulting yellow suspension stirred for a further 20 min at 5°C, prior to the addition of 4-flurophenylbromide (1.65 mmol), and the contents left to stir for a further 45 min. Saturated ammonium chloride solution was added dropwise and the contents transferred with ethyl acetate to a separating funnel whereupon the organic phase was removed, washed with water and brine, and dried over sodium sulphate. Evaporation and chromatography (SiO$_2$; gradient elution; hexane: EtOAc = 2: 1% to 100% EtOAc) afforded the desired product as an oil which was used immediately in the next reaction. The oil was added to ethanol (3 mL) and $p$-toluenesulfonic acid (53 mg, 0.27 mmol) and 4-fluorophenylaniline (0.27 mmol) at room temperature and the contents heated under reflux with stirring for 12 hr to provide a precipitate which was diluted with methanol (0.5 mL) and filtered at 50°C to afford the desired product as a pale yellow solid (45 mg, 57%). IR: v$_{max}$/cm$^{-1}$ (solid): 3130, 1615. HPLC-MS: 2.20 min, 362.4 [M+H]$^+$. $^1$H NMR (500 MHz, CDCl$_3$): δ 10.60 (d, $J$ = 12 Hz, 1H), 7.99 (d, $J$ = 12 Hz, 1H), 7.23–7.39 (m, 3H), 7.15 (d, $J$ = 9 Hz, 2H), 6.91 (m, 3H), 6.80 (d, $J$ = 8.5 Hz, 2H), 6.77 (d, $J$ = 7.9 Hz, 1H), 4.90 (s, 2H); $^{13}$C NMR (300 MHz, CDCl$_3$) 165.1, 156.5, 139.1, 138.1, 136.7, 136.1, 133.5, 132.4, 131.1, 128.9, 124.0, 122.8, 121.8, 117.0, 114.1, 111.1, 104.6, 97.5, 58.9.

## (3Z)−5-fluoro-3-[(anilino)methylidene]−1-[(4-fluorophenyl)methyl]−1,3-dihydro-2H-indol-2-one (1191−125)

A DMF (4 mL) suspension of (3Z)−3-[(dimethylamino)methylidene]−5-fluoro-1,3-dihydro-2H-indol-2-one (300 mg, 1.6 mmol) was cooled to 0°C via an ice-bath with rapid stirring. Sodium hydride (125 mg, 60% dispersion) was added in three portions over 10 min, and the resulting yellow suspension stirred for a further 20 min at 5°C, prior to the addition of 4-fluorobenzyl lbromide (1.65 mmol), and the contents left to stir for a further 45 min. Saturated ammonium chloride solution was added dropwise and the contents transferred with ethyl acetate to a separating funnel whereupon the organic phase was removed, washed with water and brine, and dried over sodium sulphate. Evaporation and chromatography (SiO$_2$; gradient elution; hexane: EtOAc = 2: 1% to 100% EtOAc) afforded an oil which was used immediately in the next reaction. The oil was added to ethanol (3 mL) and $p$-toluenesulfonic acid (53 mg, 0.27 mmol) and aniline (0.27 mmol) at room temperature and the contents heated under reflux with stirring for 12 hr to provide a precipitate which was diluted with methanol (0.5 mL) and filtered at 50°C to afford the desired product as a pale yellow solid (43 mg). IR: v$_{max}$/cm$^{-1}$ (solid): 3130, 1620. HPLC-MS: 2.30 min, 362.1 [M+H]$^+$. $^1$H NMR (500 MHz, CDCl$_3$): δ 10.65 (d, $J$ = 11.5 Hz, 1H), 7.99 (d, $J$ = 11.5 Hz, 1H), 7.15–7.39 (m, 6H), 6.91–7.12 (m, 3H), 6.77 (m, 3H), 4.90 (s, 2H).

## 4-{[(Z)-(1-(4-fluoro)phenylmethyl-2-oxo-1,2-dihydro-5-fluoro-3H-indol-3-ylidene)methyl]amino}benzonitrile (1191−126)

A DMF (4 mL) suspension of (3Z)−3-[(dimethylamino)methylidene]−5-fluoro-1,3-dihydro-2H-indol-2-one (300 mg, 1.6 mmol) was cooled to 0°C via an ice-bath with rapid stirring. Sodium hydride (125 mg, 60% dispersion) was added in three portions over 10 min, and the resulting yellow suspension stirred for a further 20 min at 5°C, prior to the addition of 4-fluorobenzylbromide (1.65 mmol), and the contents left to stir for a further 45 min. Saturated ammonium chloride solution was added dropwise and the contents transferred with ethyl acetate to a separating funnel whereupon the organic phase was removed, washed with water and brine, and dried over sodium sulphate. Evaporation and chromatography (SiO$_2$; gradient elution; hexane: EtOAc = 2: 1% to 100% EtOAc) afforded an oil which was used immediately in the next reaction. The oil was added to ethanol (3 mL) and $p$-toluenesulfonic acid (53 mg, 0.27 mmol) and aniline (0.27 mmol) at room temperature and the contents heated under reflux with stirring overnight to provide a precipitate which was diluted with methanol (2 × 0.5 mL) and filtered at 50°C to afford the desired product as a yellow solid (57 mg). IR: v$_{max}$/

cm$^{-1}$ (solid): 3100, 2200, 1615. HPLC-MS: 2.00 min, 387.1 [M+H]$^+$. $^1$H NMR (500 MHz, CDCl$_3$): δ 10.71 (d, $J$ = 12.0 Hz, 1H), 7.86 (d, $J$ = 12.0 Hz, 1H), 7.45 (br, 1H), 7.35 (d, $J$ = 8.5 Hz, 2H), 7.15 (d, $J$ = 8.5 Hz, 1H), 6.92–6.99 (m, 3H), 6.83 (m, 3H), 6.76 (t, $J$ = 7.5 Hz, 1H), 4.90 (s, 2H).

## Acknowledgements

We are grateful to Jens Bukh (Hvidovre University Hospital and University of Copenhagen, Hvidovre, Copenhagen, Denmark), Takaji Wakita (National Institute for Infectious Diseases, Tokyo, Japan) and Wendy Barclay (Imperial College, London) for the generous provision of reagents. We thank Adrian Whitehouse (Leeds) for useful discussion. We also thank Morgan Herod and Adam Davidson (Leeds) for technical advice regarding the Incucyte Zoom.

## Additional information

### Funding

| Funder | Grant reference number | Author |
|---|---|---|
| Medical Research Council | G0700124 | Matthew J Bentham<br>Laura Wetherill<br>Stephen Griffin |
| Yorkshire Cancer Research | PP025 | Toshana L Foster<br>Stephen Griffin<br>Mark Harris |
| Leeds Hospital Charitable Foundation | 9R11/14-03 | Laura Wetherill<br>Stephen Griffin |
| Medical Research Council | MC.PC.13066 | Joseph Shaw<br>Rajendra Gosein<br>Richard Foster<br>Stephen Griffin |

The funders had no role in study design, data collection and interpretation, or the decision to submit the work for publication.

### Author contributions

Joseph Shaw, Toshana L Foster, Jayakanth Kankanala, Formal analysis, Validation, Investigation, Visualization, Methodology; Rajendra Gosain, Monoj Mon Kalita, Claire Scott, Investigation, Methodology; D Ram Mahato, Barnabas J King, Emma Brown, Validation, Investigation, Methodology; Sonia Abas, Formal analysis, Investigation, Methodology; Matthew J Bentham, Laura Wetherill, Investigation; Abigail Bloy, Investigation, Writing - review and editing; Adel Samson, Writing - review and editing; Mark Harris, Resources, Investigation, Visualization, Methodology, Writing - review and editing; Jamel Mankouri, Conceptualization, Resources, Investigation, Visualization, Methodology; David J Rowlands, Conceptualization, Methodology; Andrew Macdonald, Alexander W Tarr, Conceptualization, Investigation, Methodology; Wolfgang B Fischer, Conceptualization, Resources, Formal analysis, Supervision, Validation, Investigation, Methodology; Richard Foster, Conceptualization, Resources, Formal analysis, Supervision, Funding acquisition, Validation, Investigation, Visualization, Methodology, Project administration; Stephen Griffin, Conceptualization, Formal analysis, Supervision, Funding acquisition, Investigation, Visualization, Methodology, Project administration

### Author ORCIDs

Monoj Mon Kalita http://orcid.org/0000-0002-8037-8489
D Ram Mahato https://orcid.org/0000-0002-1121-7761
Barnabas J King http://orcid.org/0000-0001-8432-2282
Alexander W Tarr http://orcid.org/0000-0003-1009-0823
Stephen Griffin https://orcid.org/0000-0002-7233-5243

### Decision letter and Author response

Decision letter https://doi.org/10.7554/eLife.52555.sa1

Author response https://doi.org/10.7554/eLife.52555.sa2

## Additional files

### Supplementary files

- Transparent reporting form

### Data availability

All data generated or analysed during this study are included in the manuscript and supporting files. Source data files have been provided for Figures.

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
