## [Decision Letter]

**Acceptance summary:**

In this work, the authors used an unbiased approach to screen for novel inhibitors of Hepatitis C Virus (HCV). They successfully identified a new HCV inhibitor which was named JK3/32. With the use of JK3/32, the authors propose a new function of the HCV p7 viroporin for viral entry into the host cell. Altogether, this study sheds light on the importance of viroporin inhibitors as new generation drug classes for HCV treatment.

**Decision letter after peer review:**

Thank you very much for submitting your article "Rationally derived inhibitors of hepatitis C virus p7 channel activity reveal prospect for bimodal antiviral therapy" for consideration by *eLife*. Your article has been reviewed by three peer reviewers, and the evaluation has been overseen by a Reviewing Editor and Olga Boudker as the Senior Editor. The following individual involved in review of your submission has agreed to reveal their identity: Gisa Gerold (Reviewer #3).

The reviewers have discussed the reviews with one another and the Reviewing Editor has drafted this decision to help you prepare a revised submission.

Summary:

In this work, the authors used an unbiased approach to screen for novel inhibitors of Hepatitis C Virus (HCV). With this strategy combined with chemical evolution to improve the initial hit compound, they successfully identified a new HCV inhibitor which was named JK3/32. The authors then computationally identified a putative binding site for this compound on the HCV p7 channel, and systematically analysed the requirement for certain functional groups for the activity of this inhibitor. With the use of JK3/32, the authors propose a new function of the HCV p7 viroporin for viral entry into the host cell. Altogether, this study sheds light on the importance of viroporin inhibitors as new generation drug classes for HCV treatment.

All three reviewers were very positive about your manuscript. They all agreed that it gives very interesting and valuable insights into HCV biology.

During our discussion with the three reviewers on your manuscript we identified two important points that we would like you to address experimentally with the highest priority in the revised version of the manuscript.

First, we would like to ask you to more directly demonstrate that p7 is indeed the target of the new inhibitor you identified by using mutational studies on p7. One way to approach this would be to mutate the amino acids responsible for JK3/32 binding (Trp48, Tyr45, Leu52).

Second, to provide experimental evidence of the genotype-specificity of JK3/32, the compound flunarazine, which specifically inhibits viral entry of the genotype 2a by targeting a hydrophobic region close to the putative E1 fusion loop (Banda et al., 2019), should be added in Figure 4C as positive control/for comparison with JK3/32. In addition, since the studies of the prototypical p7 inhibitors (Figure 4E) are done with JFH-1virus (Genotype 2a), it should also be included in Figure 4C.

Please find below the individual reviewer comments that I would like to ask you to address as well.

Reviewer 1:

The authors have performed an extensive amount of chemical, virological and in silico analysis and generated an interesting manuscript with important new findings. The issue of p7's potential role in entry is always important to consider and they have made advances in that direction. The identified compound is also interesting as a starting point for new p7 inhibitors, as the authors have pointed out. Overall this is an excellent paper that this reviewer believes is worthy of publication in this journal.

Notes on Strengths:

– The observed EC50 of 184nM for JK3/32 is significant given that it behaves similarly to SOF in compound titration assays (Figure 1B).

– JK3/32 did not affect replicons.

– The lack of effect on IAV is convincing regarding target specificity.

– The labelled tool provides a useful reagent for multiple types of future work.

I have some concern with Figure 4A. Is the effect on entry dose-dependent? The authors have presented only 0 and 4uM tests on entry and although the data is statistically significant, should we consider an ~2.5-fold decrease in entry as biologically significant? There is also some concern that effects were only seen at μM compound concentrations (Figure 4A and C). Additional concern is also raised by the fact that the authors report they were not able to generate resistance to the compound after multiple attempts. They have discussed this in the subsection “ HCV entry is dependent upon p7 ion channel function”, and attributed it to sequence conservation and perhaps necessity in the compound binding site.

The data showing that pseudotyped viruses were not affected by the compound (Figure 4F) does indeed suggest that p7 is involved in entry and is then inhibited by the compound in the context of HCVcc. However, how can the authors be sure that the p7 function during virus assembly/release that would protects E1E2 from premature degradation is not inhibited by the compound and then indirectly affecting entry because released particles would have deficient envelopes? In this case, the current pattern of results would remain – one would expect to see an inhibition of HCVcc entry but not HCVpp. So does this data really suggest p7 is active during entry? This is not easy to discern experimentally, but perhaps the authors might discuss this possibility in the context of premature degradation of E1E2.

The demonstration of p7 being present within the virion would be the conclusive evidence that has been sought after for years to prove p7 might be involved in entry. The authors are well aware of this and have addressed is as well as anyone could do. The future experiments discussed in the eleventh paragraph of the Discussion will be innovative approaches to readdress this issue.

Reviewer 2:

In Shaw et al. the authors describe an unbiased approach for screening novel inhibitors of HCV. Using this strategy followed by chemical evolution to improve the initial hit, they successfully identify a new inhibitor: JK3/32. The authors were then able to computationally identify a putative binding site for the molecule on the HCV p7 channel, and systematically test the requirement of several functional groups for inhibitor activity. Using JK3/32, the authors suggest a new function of p7 in viral entry.

1) One question I cannot answer is whether the identified compound is different enough from previously identified drugs to represent a substantial breakthrough from what is already known.

2) Figure 1—figure supplement 2: Is the increase in activity of some human kinases statistically significant? Have the authors considered whether an increase in activity could also lead to toxicity in cells?

3) Figures 2/3: Is it possible to confirm the binding site of JK3/32 to p7 directly using mutational analysis to strengthen the computational and simulation analysis reported here? It seems the authors do something similar in Atkins et al., 2014? Perhaps the effects of JK3/32 on different viral genotypes could shed some light on this, provided viruses containing natural mutations within the binding site have already been identified.

4) Figure 4: As with above, can it be demonstrated that direct mutations to p7 affect viral entry? Experiments described in Figure 5 convincingly demonstrate that the effect of JK3/32 is unlikely an off-target effect on the host cells, but it would be helpful to more directly establish that this effect is due to binding only to the p7 channel. This is particularly true in light of the apparent difficulty of providing evidence that p7 channels exist directly in virions.

Reviewer 3:

The manuscript by Shaw et al. identifies a potent HCV p7 inhibitor (p7i) by structure-activity relationship (SAR) studies. In addition, rational SAR-based compound design in combination with atomistic molecular dynamics revealed the basic compound structure for their inhibition studies and pinpointed the potential binding pocket of JK3/32. Interestingly, the authors could show that JK3/32 is not only inhibiting viral secretion but also interferes with HCV cell entry. This study is well designed and performed. Moreover, it supports the role of p7 as an important viral protein for HCV entry. Altogether this study brings to light the importance of viroporin inhibitors as new generation drug classes for HCV treatment.

The following suggestions would increase the overall clarity of the study and improve the quality of the data.

1) The molecular dynamics simulation studies would be benefit from the addition of the RMSD values of the unbound protein (without JK3/32) for comparison

2) A detailed scheme of the molecular interactions of JK3/32 with Leu20Phe p7 mutant as for 3B would be a nice addition to the current Figure 3C.

3) Entry inhibition studies (Figure 4) would benefit from a positive control drug such as flunarizine, which targets the potential fusion peptide within E1.

4) The HCVpp experiments with prototypical glycoproteins (Figure 4—figure supplement 1C) and VSV-Gpps are missing the GNA control.

---

## [Author Response]

All three reviewers were very positive about your manuscript. They all agreed that it gives very interesting and valuable insights into HCV biology.During our discussion with the three reviewers on your manuscript we identified two important points that we would like you to address experimentally with the highest priority in the revised version of the manuscript.First, we would like to ask you to more directly demonstrate that p7 is indeed the target of the new inhibitor you identified by using mutational studies on p7. One way to approach this would be to mutate the amino acids responsible for JK3/32 binding (Trp48, Tyr45, Leu52).

We naturally agree that this question is of major importance and in fact alludes to three overlapping aspects of the work, namely:

1) Specific interactions between JK3/32 and the (genotype 1b) p7 channel complex

2) The SAR for our inhibitor series relating to the peripheral binding site, extending to other HCV genotypes

3) The functional role for p7 channel activity during virus entry

The pandemic lockdown has frustrated our attempts to conduct experiments that directly address point one, yet we have now included data that we feel convincingly address the other two points and offers insight around the first.

Selection of a resistance polymorphism would provide excellent evidence for JK3/32 target engagement across all three areas listed above. In this regard, we undertook multiple cell culture experiments where we attempted to select JK3/32 resistance by sequential passage with increasing inhibitor concentration, and were unable to detect changes by deep sequencing even prior to elimination of the virus from culture (even at just a single EC_50_). This was not rescuable by subsequent blind passage in the absence of drug. This led to our conclusion that the invariant nature of the binding site residues prevented resistance from arising. However, it is also possible that a single substitution is not sufficient for resistance to evolve.

Mutation of key interacting residues such as W48 or L52 is certainly one approach to disrupting JK3/32 binding. However, these are 100% conserved within naturally occurring p7 sequences, suggesting important functional roles. Accordingly, a W48F mutation (and Y42F) in a genotype 2a JFH-1 background caused significant defects in both virion infectivity and core protein secretion (Steinmann et al., 2007).

Previously, we identified the F25A imino-sugar resistance polymorphism via inter-genotypic variation within the predicted binding site in molecular models (Foster et al., 2011). However, JK3/32 displays variable activity against the three HCV genotypes included in this study rather than discrete resistance, making it possible, or even likely, that individual mutations might not give a clear picture of JK3/32 binding.

Thus, an inhibitor with a discrete resistance phenotype would represent a better means of validating the peripheral binding site and the role of p7 during virus entry. Whilst this was shown for rimantadine and the L20F polymorphism (Figure 4D), a member of the JK3/32 inhibitor series would be more relevant to future development.

Frustratingly, we had overlooked the existence of such a compound due to its use as a negative control for the majority of our genotype 1b experiments, namely R21. Unlike genotype 1b and 3a, the R21 compound displays moderate efficacy against the JFH-1 strain (EC_50_=4.1 μM vs. secreted infectivity, revised Figure 5A). This led to us not including JFH-1 in the JK3/32 data presented in Figure 4, as an appropriate negative control was not available; the reviewers rightly noted this exception. However, R21 did indeed inhibit JFH-1 entry, supporting that JK3/32 series inhibitors targeting the peripheral binding site block an important aspect of HCV entry mediated by p7 channel activity (revised Figure 5B).

As implied above, genotypic variation within the JK3/32 series is likely due to both the composition of key residues in the peripheral binding site as well as the context of the overall channel structure. For example, the L20F rimantadine resistance polymorphism increased susceptibility of JFH-1 to certain compounds from our first hit series, whilst it reduced sensitivity for genotype 1b (Foster et al., 2014).

Thus, we hypothesised that variation in the JFH-1 binding site may not only explain R21 sensitivity, but may also provide insight into potential resistance mechanisms for other genotypes and members of the SAR series, including JK3/32. As shown in revised Figure 5, aside from L20, only positions Y45, L50 and L51 (genotype 1b sequence) display limited variability within the 1b, 2a and 3a binding sites. Compared to J4, JFH-1 (genotype 2a) shows a Y45T/L50F/L51C change, whilst S52 (genotype 3a) contains Y45T/L51A. We had engineered several of these mutations into the J4/JFH-1 virus background prior to the COVID19 lockdown, albeit with some trepidation regarding their potential effects upon viral fitness, as discussed above.

Using established p7 structural homology models (Foster et al., 2014), we examined predicted interactions between R21 and the genotype 1b, 2a and 3a binding sites (revised Figure 5C). Y45T and L50F in JFH-1 (position 51 is likely too distant to directly influence interactions) are predicted to influence the way in which R21 interacts with the peripheral binding site, generating a deeper binding pocket that better incorporates the R21 structure. Y45T also disrupts an interaction between Y45 and Y42 present in the 1b channel model, allowing the “northern” R21 dimethoxy-phenyl ring to better-fit within the cavity. In turn, L50F allows the “southern” phenyl ring to fit within the pocket without the distortion seen in other genotypes; both mutations result in significantly reduced exposure of R21 to the surrounding environment. We infer that both Y45T and L50F are required for R21 sensitivity as the resistant genotype 3a channel lacks the latter change.

We next generated genotype 1b channel models containing either Y45T/L50F or Y45T/L51C to assess whether Y45T, L50F or both might alter predicted drug binding (revised Figure 6). Interestingly, both mutations reduced the predicted interaction efficiency with JK3/32, suggesting that Y45T underpins the decreased susceptibility of JFH-1 and S52 channels to the compound but that this does not itself constitute a resistance polymorphism. However, introducing the double mutations into the genotype 1b background did not increase the predicted binding efficiency of R21.

Thus, we conclude that resistance to JK3/32 may potentially arise via the Y45T mutation, but this would also require either additional adjacent changes to the binding site, or potentially further alterations to the global channel structure achieved via more distal amino acid changes. Interestingly, it seems likely that compounds evolved from R21 might therefore be used to counter resistance emerging to JK3/32. Taken together, we believe these observations explain our failure to select JK3/32 resistance in culture, suggest that single or even double mutations in the J4 sequence as suggested may not serve as an alternative, and yet also provide orthogonal validation of the peripheral binding site as well as the requirement for p7 during HCV entry.

Second, to provide experimental evidence of the genotype-specificity of JK3/32, the compound flunarazine, which specifically inhibits viral entry of the genotype 2a by targeting a hydrophobic region close to the putative E1 fusion loop (Banda et al., 2019), should be added in Figure 4C as positive control/for comparison with JK3/32. In addition, since the studies of the prototypical p7 inhibitors (Figure 4E) are done with JFH-1virus (Genotype 2a), it should also be included in Figure 4C.

We are happy to conduct flunarazine experiments, as requested, as soon as we are able (see above). However, we infer that this experiment is a means of confirming that an additional class of inhibitor is capable of preventing HCV entry. In this regard, we point out that our revised manuscript now shows data on four structurally distinct p7 inhibitor compounds that prevent HCV entry (rimantadine, *N*N-DNJ, JK3/32 and R21), with the L20F and other (predicted) polymorphisms confirming site and class specificity (i.e. L20F prevents rimantadine, but not *N*NDNJ activity; JFH-1 susceptible to R21 etc.). Moreover, GNA also specifically prevents the entry of HCV-pseudotyped Lentiviruses compared to those coated with the VSV-G protein (Figure 4F, Figure 4—figure supplement 1) and we have provided numerous controls for effects upon cellular processes and global virus uptake.

Please find below the individual reviewer comments that I would like to ask you to address as well.Reviewer 1:The authors have performed an extensive amount of chemical, virological and in silico analysis and generated an interesting manuscript with important new findings. The issue of p7's potential role in entry is always important to consider and they have made advances in that direction. The identified compound is also interesting as a starting point for new p7 inhibitors, as the authors have pointed out. Overall this is an excellent paper that this reviewer believes is worthy of publication in this journal.Notes on Strengths:– The observed EC50 of 184nM for JK3/32 is significant given that it behaves similarly to SOF in compound titration assays (Figure 1B).– JK3/32 did not affect replicons.– The lack of effect on IAV is convincing regarding target specificity.– The labelled tool provides a useful reagent for multiple types of future work.I have some concern with Figure 4A. Is the effect on entry dose-dependent? The authors have presented only 0 and 4uM tests on entry and although the data is statistically significant, should we consider an ~2.5-fold decrease in entry as biologically significant? There is also some concern that effects were only seen at μM compound concentrations (Figure 4A and 4C).

The reviewer is correct that Figure 4A shows a single concentration, yet Figure 4C contains titrations of JK3/32 and R21 for both J4/JFH-1 and S52/JFH-1 chimaeric viruses, as revised Figure 5 does for JFH-1.

The point regarding the relevance of the observed fold-reduction is difficult to answer and may relate to the discussion regarding saturation of virion-resident channels in our original submission. Nevertheless, the effect *is* dose dependent and the experiment in the original Figure 5 (now revised Figure 7) where direct treatment of purified virus with 10 μM JK3/32 caused a drastic reduction in titre, suggests that saturation is achievable above a threshold drug concentration.

We naturally shared the concerns of the reviewer regarding the higher concentrations required, as discussed extensively within the manuscript, and these drove us to undertake numerous control experiments.

We also highlight that four different p7 inhibitors herein exert dose-dependent effects upon virus entry in either a polymorphism (L20F), or genotype-dependent fashion, namely: rimantadine, imino-sugars, JK3/32 and R21.

Additional concern is also raised by the fact that the authors report they were not able to generate resistance to the compound after multiple attempts. They have discussed this in the subsection “ HCV entry is dependent upon p7 ion channel function”, and attributed it to sequence conservation and perhaps necessity in the compound binding site.

Please see discussion and description of new data above. We have no reason to doubt that a high genetic barrier exists to JK3/32 resistance exists based upon the high degree of conservation within the binding site. However, the polymorphism-dependent actions of rimantadine and the R21 compound support the specificity of the peripheral binding site, and the relevance of p7 channel activity during entry. We regret that we have been unable to assess our HCV mutants due to the ongoing pandemic, albeit with fairly low confidence that a definitive phenotype might have been achieved.

The data showing that pseudotyped viruses were not affected by the compound (Figure 4F) does indeed suggest that p7 is involved in entry and is then inhibited by the compound in the context of HCVcc. However, how can the authors be sure that the p7 function during virus assembly/release that would protects E1E2 from premature degradation is not inhibited by the compound and then indirectly affecting entry because released particles would have deficient envelopes?

Apologies for any confusion. The experiments involving Lentiviral pseudotypes involved the addition of inhibitors during entry, rather than during particle production. We have clarified this in the text. Please see below regarding discussion of p7 function during secretion.

In this case, the current pattern of results would remain – one would expect to see an inhibition of HCVcc entry but not HCVpp. So does this data really suggest p7 is active during entry? This is not easy to discern experimentally, but perhaps the authors might discuss this possibility in the context of premature degradation of E1E2.

As above, inhibitors are not present during secretion in these experiments and we have further clarified the relevant text.

Whilst we have shown that p7 channel activity functions during secretion to protect the functionality of E1 and E2, we do not attribute this to premature degradation of these proteins. We do not see a reduction of E2 within inhibitor treated cells, for example (Foster et al., 2011); however, amantadine and *N*NDNJ can alter glycosylation patterns in both viral and cellular proteins (Griffin et al., 2008).

Where E1/E2 degradation does occur is in the context of p7 point mutations, such as those involving the conserved basic loop residues. However, evidence points to degradation of the precursor E2-p7-NS2 complex, rather than the glycoproteins in isolation (Bentham et al., 2014). We hypothesise (as described in the Discussion during description of specific infectivity measurements) that p7 may insert into the membrane post-translationally, independent of the SRP – this explains how HCV with deletion of half of the p7 sequence remains viable despite predicted effects upon polyprotein topology (papers from Thomas Pietschmann laboratory). Hence, basic loop mutations and some other changes seemingly cause E2 degradation because of aberrant E2-p7-NS2 processing.

Notably, the majority of intracellular particles are associated with the ER (Gastaminza et al., 2007) and so are not exposed to acidification that might be expected to induce glycoprotein degradation; p7 inhibitors therefore have little effect upon bulk intracellular infectivity and solely block secretion of infectious virus (Foster et al., 2011). Moreover, they are seemingly less effective against cell-to-cell spread, which presumably bypasses the acidifying secretory pathway followed by released virions (Meredith et al., 2013).

The demonstration of p7 being present within the virion would be the conclusive evidence that has been sought after for years to prove p7 might be involved in entry. The authors are well aware of this and have addressed is as well as anyone could do. The future experiments discussed in the eleventh paragraph of the Discussion will be innovative approaches to readdress this issue.

We wholly agree. Our previously published p7-specific antisera were previously employed in this regard (EM, antibody-mediated blockade of infection), yet the data were not reproduced to a satisfactory standard prior to supplies of our sera being sadly exhausted. Nevertheless, whilst preliminary data supported the presence of p7 within virions, we are keen to confirm this using our modified compounds.

Reviewer 2:In Shaw et al. the authors describe an unbiased approach for screening novel inhibitors of HCV. Using this strategy followed by chemical evolution to improve the initial hit, they successfully identify a new inhibitor: JK3/32. The authors were then able to computationally identify a putative binding site for the molecule on the HCV p7 channel, and systematically test the requirement of several functional groups for inhibitor activity. Using JK3/32, the authors suggest a new function of p7 in viral entry.1) One question I cannot answer is whether the identified compound is different enough from previously identified drugs to represent a substantial breakthrough from what is already known.

We assume that the reviewer refers to the first generation hit compounds (e.g. LDS19, Figure 1—figure supplement 3), rather than prototypic inhibitors such as rimantadine etc. The first gen compounds were screened in silico using our NMR-based channel structure template and some, including LDS19, showed similar potency to JK3/32 (Foster et al., 2014). However, LDS19 was one of a number of structurally distinct hits lacking a common pharmacophore, meaning that we were unable to derive an SAR.

The iterative development of LDS19 into JK3/32 and other structural analogues allowed us to correlate the functionally important aspects of the molecule with the predicted structure, generating a comprehensive SAR. In turn, this alignment of chemical variants with relative potency/lack thereof provided us with confidence around the biological relevance of our hairpin-based channel model, in preference to the genotype 5a structure. Moreover, we were able to predict and subsequently prove that e.g. R21 is unlikely to demonstrate potency vs. GT1b channels as well deriving the modified tool compound. Without this chemical and structural information, we would not have possessed the requisite tools to investigate the controversial role of p7 during HCV entry.

Thus, JK3/32 is clearly superior to prototypic p7i and BIT225 in terms of potency and specificity, yet is also a major leap forwards in the chemical toolbox with which to investigate fundamental biology, as well as taking forward for future drug development.

2) Figure 1—figure supplement 2: Is the increase in activity of some human kinases statistically significant? Have the authors considered whether an increase in activity could also lead to toxicity in cells?

We apologise for not explaining this assay sufficiently. The data shown are an average of two in vitro measurements assessing recombinant/purified kinases activity in the presence of drug or carrier; we have added the appropriate error bars for standard deviations. This is a well-regarded service provided by the MRC PPU in Dundee, albeit capturing only a subset of the kinome expressed within Huh7 or other cells.

The reviewer rightly notes the apparent increase in activity for certain kinases. No statistically significant differences were discernible across the data set using multiple T-Tests, and the relevance of increased activity within an in vitro setting is not clear. Moreover, addition of 10-μM compound to native kinases presumably reflects a much higher level exposure than that likely to occur within a cell, even when using the JK3/32 concentrations required to inhibit virus entry.

Detailed exploration of potential off-target effects upon cellular kinases would comprise part of future drug development, yet the kinase screen herein constitutes just one aspect of the toxicity screening undertaken during this study. Whilst some compounds were indeed found to adversely affect cellular viability, JK3/32 and the majority of the related series showed encouraging selectivity indices, based upon both producer cell confluency (IncuCyte) and metabolic activity (MTT).

In addition, we tested several compounds for effects upon cellular confluency over time, again using the IncuCyte. The additional data, shown in Author response image 1, shows negligible effects for JK3/32, whereas BIT225 (potently) and rimantadine (high concentrations) do indeed affect this process. We can incorporate this data into the manuscript if required, although not all of the series was assessed in this way – please do let us know.

**Author response image 1. sa2fig1:** 

3) Figures 2/3: Is it possible to confirm the binding site of JK3/32 to p7 directly using mutational analysis to strengthen the computational and simulation analysis reported here? It seems the authors do something similar in Atkins et al., 2014? Perhaps the effects of JK3/32 on different viral genotypes could shed some light on this, provided viruses containing natural mutations within the binding site have already been identified.

Please see above for discussion of this very important aspect of the study.

The sequences studied in Atkins et al., 2014 were complete p7 sequences cloned from a genotype 1a patient cohort, then inserted into a H77/JFH-1 infectious clone backbone. Thus, we had no direct input into the nature of these mutations. Notably, several mutations caused variable effects upon viral “fitness” in terms of protein expression, infectious virus production, and indeed inhibitor sensitivity. This is perhaps unsurprising given studies from e.g. the Bartenschlager laboratory showing complementation of p7 mutations by changes in NS2, and other viral proteins.

4) Figure 4: As with above, can it be demonstrated that direct mutations to p7 affect viral entry? Experiments described in Figure 5 convincingly demonstrate that the effect of JK3/32 is unlikely an off-target effect on the host cells, but it would be helpful to more directly establish that this effect is due to binding only to the p7 channel. This is particularly true in light of the apparent difficulty of providing evidence that p7 channels exist directly in virions.

We agree that this would be an ideal experiment, yet it is unfortunately the case that mutations intended to disrupt p7 ion channel function have deleterious effects upon alternative functions. Extensive discussion of the most commonly employed mutation in this regard, substitutions of the cytosolic basic loop, is already present within the manuscript.

We have undertaken extensive mutagenesis of the p7 sequence in several HCV genotypes, analysing phenotypes both in vitro and in the context of virus infection. Often mutations have effects upon E2-p7-NS2 processing/stability, as discussed in the response to reviewer 1. As described above, it remains to be seen whether rational mutagenesis of the peripheral binding site might attenuate JK3/32 binding in the J4 p7 context.

Finally, returning to the three points raised at the start of this rebuttal, we infer that the reviewer appears more interested in determining genetic evidence for the role of p7 during entry, rather than the specific interactions underpinning JK3/32 binding. We again note that a single amino acid change in the p7 sequence specifically alters the sensitivity of HCV to a p7 inhibitor, namely rimantadine. Moreover, if our predictions are correct then the action of R21 upon JFH-1 entry is similarly supportive. Thus, genetic evidence of a requirement for p7 channel activity during entry already – in our view – exists within the data presented herein, yet we obviously regret not being able to conduct further experiments to clarify this.

Reviewer 3:The manuscript by Shaw et al. identifies a potent HCV p7 inhibitor (p7i) by structure-activity relationship (SAR) studies. In addition, rational SAR-based compound design in combination with atomistic molecular dynamics revealed the basic compound structure for their inhibition studies and pinpointed the potential binding pocket of JK3/32. Interestingly, the authors could show that JK3/32 is not only inhibiting viral secretion but also interferes with HCV cell entry. This study is well designed and performed. Moreover, it supports the role of p7 as an important viral protein for HCV entry. Altogether this study brings to light the importance of viroporin inhibitors as new generation drug classes for HCV treatment.The following suggestions would increase the overall clarity of the study and improve the quality of the data.1) The molecular dynamics simulation studies would be benefit from the addition of the RMSD values of the unbound protein (without JK3/32) for comparison

Apologies, this was only included in supplementary figures for rimantadine previously – this is now rectified.

2) A detailed scheme of the molecular interactions of JK3/32 with Leu20Phe p7 mutant as for Figure 3B would be a nice addition to the current Figure 3C.

We entirely agree, and this has now been included, although the Leu20Phe mutation data is now moved to Figure 3—figure supplements 1 and 2 due to space constraints.

3) Entry inhibition studies (Figure 4) would benefit from a positive control drug such as flunarizine, which targets the potential fusion peptide within E1.

Please see above discussion. Again, we are happy to conduct such experiments when feasible, but also feel that the accumulation of genotype and compound-specific activity during entry is similarly supportive.

4) The HCVpp experiments with prototypical glycoproteins (Figure 4—figure supplement 1C) and VSV-Gpps are missing the GNA control.

Apologies, this is now included showing HCV-specific effects with no effect upon VSV-G pseudotyped viruses.